# Ca$^{2+}$-modulated photoactivatable imaging reveals neuron-astrocyte glutamatergic circuitries within the nucleus accumbens

Irene Serra[1], Julio Esparza [1], Laura Delgado[1], Cristina Martín-Monteagudo [1], Margalida Puigròs[2], Petar Podlesniy[2], Ramón Trullás [2] & Marta Navarrete [1]✉

Astrocytes are key elements of brain circuits that are involved in different aspects of the neuronal physiology relevant to brain functions. Although much effort is being made to understand how the biology of astrocytes affects brain circuits, astrocytic network heterogeneity and plasticity is still poorly defined. Here, we have combined structural and functional imaging of astrocyte activity recorded in mice using the Ca$^{2+}$-modulated photoactivatable ratiometric integrator and specific optostimulation of glutamatergic pathways to map the functional neuron-astrocyte circuitries in the nucleus accumbens (NAc). We showed pathway-specific astrocytic responses induced by selective optostimulation of main inputs from the prefrontal cortex, basolateral amygdala, and ventral hippocampus. Furthermore, co-stimulation of glutamatergic pathways induced non-linear Ca$^{2+}$-signaling integration, revealing integrative properties of NAc astrocytes. All these results demonstrate the existence of specific neuron-astrocyte circuits in the NAc, providing an insight to the understanding of how the NAc integrates information.

Although astrocytes have been traditionally regarded as a homogeneous population of cells, accumulating evidence indicates that there is functional heterogeneity in the astrocyte subpopulations[1–4] from different brain regions based on their morphology, functionality, physiological properties, developmental origin and response to pathologies[5–8]. Neuron–astrocyte communication has been shown to be a key element in brain physiology, yet the heterogeneity in the functional interaction between astrocytes and synapses in different brain circuitries remains poorly understood.

The nucleus accumbens (NAc) is a part of the basal forebrain that plays a relevant role in the reward system in numerous neurological and psychiatric disorders[9]. One remarkable feature of NAc is that it receives and integrates different glutamatergic signals from several brain regions, including the medial prefrontal cortex (mPFC), basolateral amygdala (Amyg), and ventral hippocampus (vHip). The NAc has been divided into two different structures, the so-called shell (AcbSh) and core (AcbC), distinguished by differential expression of neuropeptides, morphology, membrane properties, and synaptic inputs from afferent structures. The AcbC is related to guiding behavior towards a specific goal based on learning, whereas the AcbSh is involved in unconditioned reward-seeking behaviors[10].

The particular pathway-specific activation of glutamatergic fibers to the NAc has been demonstrated to evoke diverse physiological and behavioral responses[10,11]. For example, the mPFC-NAc pathway interacts with behaviors that require not only visual attention, but also attention in action, taking into account the outcomes of expected/anticipated reward[12,13], working memory[14] or the integration of different inputs[15]. vHip afferents regulate rewards related to contextual cues or reward-related memories[15–17], which are relevant to disorders that alter reward processing, e.g., addiction or anhedonia, and are involved in susceptibility to stress disorders[16,18]. Lastly, afferents from Amyg are involved in value and emotional decisions, and reward-seeking, focused on positive valence[19–21]. However, all these studies were centered mainly on the innervation patterns of distinct input and/or output features of different types of neurons[9,22,23], leaving the functional intersection of astrocytes in these hallmarks underappreciated for decades.

[1]Instituto Cajal, CSIC, Madrid, Spain. [2]Institute for Biomedical Research of Barcelona, CSIC, CIBERNED, Barcelona, Spain. ✉e-mail: mllinas@cajal.csic.es

Considering astrocytes as integrative space-time detectors of neuronal signaling[24,25], we explored whether astrocytes establish segregated neuron−astrocyte networks within the NAc that have intrinsic properties and functional consequences to the neuronal circuit´s output. To this end, we optogenetically manipulated the specific synaptic afferents to the NAc in combination with a new adapted technique, calcium-modulated photoactivatable ratiometric integrator under GFAP promoter (CaMPARI$_{GFAP}$), to selectively dissect Ca$^{2+}$ astrocyte signaling. We demonstrated that selective optostimulation of main glutamatergic inputs (i.e., mPFC, Amyg, and vHip) induces astrocytic Ca$^{2+}$ activities mediated by mGluR5 that do not coincide with glutamatergic innervation patterns, suggesting unexpected neuron−astrocyte circuitries.

## Results

### CaMPARI$_{GFAP}$ as a functional tool to monitor astrocyte activity

To study functional neuron−astrocyte circuitries in the NAc, we used a fluorescent technique based on CaMPARI1[26,27] a genetically encoded Ca$^{2+}$ indicator (GECI) expressed in astrocytes that undergoes irreversible green-to-red fluorescence conversion upon coincident elevated intracellular Ca$^{2+}$ and violet ($\lambda = 405$ nm) light illumination[27]. The imaging of CaMPARI1 green fluorescence allows real-time monitoring of astrocytic Ca$^{2+}$ dynamics, while its irreversible photoconversion properties (red fluorescence) enable a large-scale spatial analysis of astrocytic activation with precise temporal resolution[26–28]. We selectively expressed CaMPARI1[27] in NAc astrocytes by injecting the adeno-associated virus pAAV2/9.GFAP.CaMPARI.WPRE.SV40 (CaMPARI$_{GFAP}$), which contains the specific astroglial promoter glial fibrillary acidic protein (GFAP) in the NAc (Fig. 1A). Specific expression of CaMPARI$_{GFAP}$ in NAc astrocytes was confirmed for both green and red fluorescence signals by its colocalization with astrocytic marker S100, but negligibly with the neuronal marker NeuN (95.7 ± 0.7% of CaMPARI$_{GFAP}$ green cells and 94.1 ± 1.7% of CaMPARI$_{GFAP}$ red cells were S100-positive) ($P < 0.001$; $n = 2861$ cells, 8 fields, 2 mice) (Fig. 1B). Afterward, we studied both the ability to track real-time Ca$^{2+}$ astrocytic activity and photoconversion properties[27]. Using the local application of ATP (20 mM) through a micropipette, which reliably elevates intracellular Ca$^{2+}$ levels in NAc astrocytes[29], we showed a transient decrease in CaMPARI$_{GFAP}$ green fluorescence in astrocytes (Fig. 1C–F)[30,31]. This change in the green fluorescence signal was prevented by perfusing 1 μM thapsigargin (1.67 ± 1.67% in thapsigargin vs 98.9 ± 1.11% in control; $n = 6$ and $n = 9$ slices, in thapsigargin and control conditions respectively; $P < 0.001$) which depletes the internal stores by inhibiting the Ca$^{2+}$ ATPase. The same was achieved by loading BAPTA (20 mM) into the astrocytes syncytium using a patch pipette ($P < 0.001$; Fig. 1F, see Supplementary Fig. S1 for a representative example of astrocyte intracellular loading with BAPTA and biocytin, followed by streptavidin-Alexa 647 staining). It is well known that BAPTA spreads via gap junctions into the astrocyte syncytium network, blocking astrocytic Ca$^{2+}$ signaling throughout the slice[32–37]. These results demonstrate the viability of the molecule to monitor Ca$^{2+}$ dynamics in astrocytes.

In parallel, the application of 405 nm light during a fixed temporal window (10 s) right after ATP local delivery led to green-to-red photoconversion of activated astrocytes (Fig. 1G); to note, no tissue damage was detected due to 405 nm light application[38–40] (Supplementary Fig. S2, see "Methods"). To cover large areas of tissue, CaMPARI$_{GFAP}$ green and red fluorescences were measured post hoc after PFA fixation. As shown in Fig. 1H, the CaMPARI$_{GFAP}$ red and green fluorescences (ΔF/F0) changed according to the distance from ATP application. These fluorescence changes were also analyzed in the presence of thapsigargin or BAPTA infusion, revealing no photoconversion nearby the stimulus, which shows that the green-to-red photoconversion ratio correlated with Ca$^{2+}$ activity (red fluorescence is increased at closer distances to the micropipette while green fluorescence is decreased).

Although some studies report that CaMPARI1 fluorescence signal is reduced upon chemical fixation[27,28], photoconversion was robust in astrocytes, and this decrease did not compromise CaMPARI$_{GFAP}$ post hoc signal detection (Fig. 1D). Furthermore, no significant photoconversion was detected when tissue was not exposed to 405 nm light, confirming that background CaMPARI$_{GFAP}$ spontaneous photoconversion is residual and does not affect the reported measurements (Supplementary Fig. S3). Overall, these observations indicate that astrocytic Ca$^{2+}$ activity can be studied using both CaMPARI$_{GFAP}$ properties: real-time monitoring of calcium dynamics, and large-scale activation profiles given by fluorescence irreversible photoconversion.

### Optostimulation of mPFC axons induces specific activation of a subpopulation of astrocytes in the NAc

To study neuron−astrocytic circuitry heterogeneity in the NAc, we first analyzed the neural element of the circuit, characterizing the anatomical and functional patterns of the mPFC glutamatergic afferents to the NAc (Fig. 2A–E). To this end, we injected AAVs expressing the opsins channelrhodopsin (ChR2) or ChrimsonR, with mCherry or tdTom fluorescent reporters (AAV5-CaMKIIa-hChR2(H134R)-mCherry; AAV5-hSync-ChrimsonR-tdTom) in the mPFC. Four weeks after injection, we analyzed the projecting fluorescence signals from mPFC axons specifically in the NAc. AcbC and AcbSh subregions were analyzed to characterize both the density and synaptic strength of mPFC afferent inputs. In agreement with previous studies[41,42], we found a non-uniform distribution of mPFC afferents in the NAc, with higher innervation density in the AcbC (0.25 ± 0.04 arb.u.) than the AcbSh (0.14 ± 0.03 arb.u.) ($P = 0.02$; $n = 11$ infections, 6 mice; Fig. 2B). Optogenetic stimulation of the mPFC with either ChR2 or ChrimsonR (Supplementary Fig. S4) evoked glutamatergic excitatory postsynaptic currents (EPSCs; as indicated by their sensitivity to 20 μM CNXQ and 50 μM AP5; Supplementary Fig. S4D), confirming the possibility to use them equally as stimulation systems. The amplitude of EPSCs recorded in AcbC (208.6 ± 37.4 pA; $n = 8$ cells, 4 mice) and AcbSh neurons (33.1 ± 12.3; $n = 15$, 4 mice: $P = 0.002$; Fig. 2E) correlated positively with the afferent innervation density (Pearson $r = 0.73$, $P < 0.001$; Fig. 2D). These results suggest that mPFC neurons preferentially innervate the AcbC subregion and that the opto-evoked EPSC amplitude in NAc neurons was dependent on the density of innervating glutamatergic fibers.

We next investigated the astrocytic Ca$^{2+}$ dynamics associated with the mPFC glutamatergic pathway (Fig. 2F–H), focusing on the postsynaptic element in the neuron−astrocyte circuitry under study. To this end, we monitored in both NAc subregions the Ca$^{2+}$ events using real-time CaMPARI$_{GFAP}$ green fluorescence signal, before and after optogenetic stimulation (10 pulses at 4 Hz, 200 ms interval−four times, 5 s interval). AAV5-hSync-ChrimsonR-tdTom was used for optogenetic stimulation in order to be able to perform imaging recordings in the blue light spectrum without interfering with the afferents. This protocol increased the astrocyte Ca$^{2+}$ spike frequency in both the AcbC (1.62 ± 0.23 change from basal, $P = 0.02$) and AcbSh (2.62 ± 0.52 change from basal, $P = 0.003$) (2420 ROIs; $n = 25$ slices, 8 mice; Fig. 2H, right) without variations in the Ca$^{2+}$ amplitude of the responses (Supplementary Fig. S5A). Surprisingly, these Ca$^{2+}$ dynamics in response to optostimulation did not show the same pattern of activity across NAc regions as observed by the glutamatergic inputs' innervation, indicating that astrocytes were not responding exclusively in areas within the direct reach of the glutamatergic afferents. Although the AcbC showed a higher density of mPFC projections, higher astrocytic activity was detected in the AcbSh (Fig. 2B, H). This increase in Ca$^{2+}$ response was mediated by group I metabotropic glutamate receptor in both AcbC and AcbSh subregions, as when it was blocked by the selective mGluR5 antagonist MPEP (50 μM) (0.87 ± 0.14 change from basal, $P = 0.42$; $n = 4$ slices, 2 mice; Fig. 2H, left). Overall, these results showed that mPFC inputs induce astrocytic activity in a pattern across subregions different from the innervation profile.

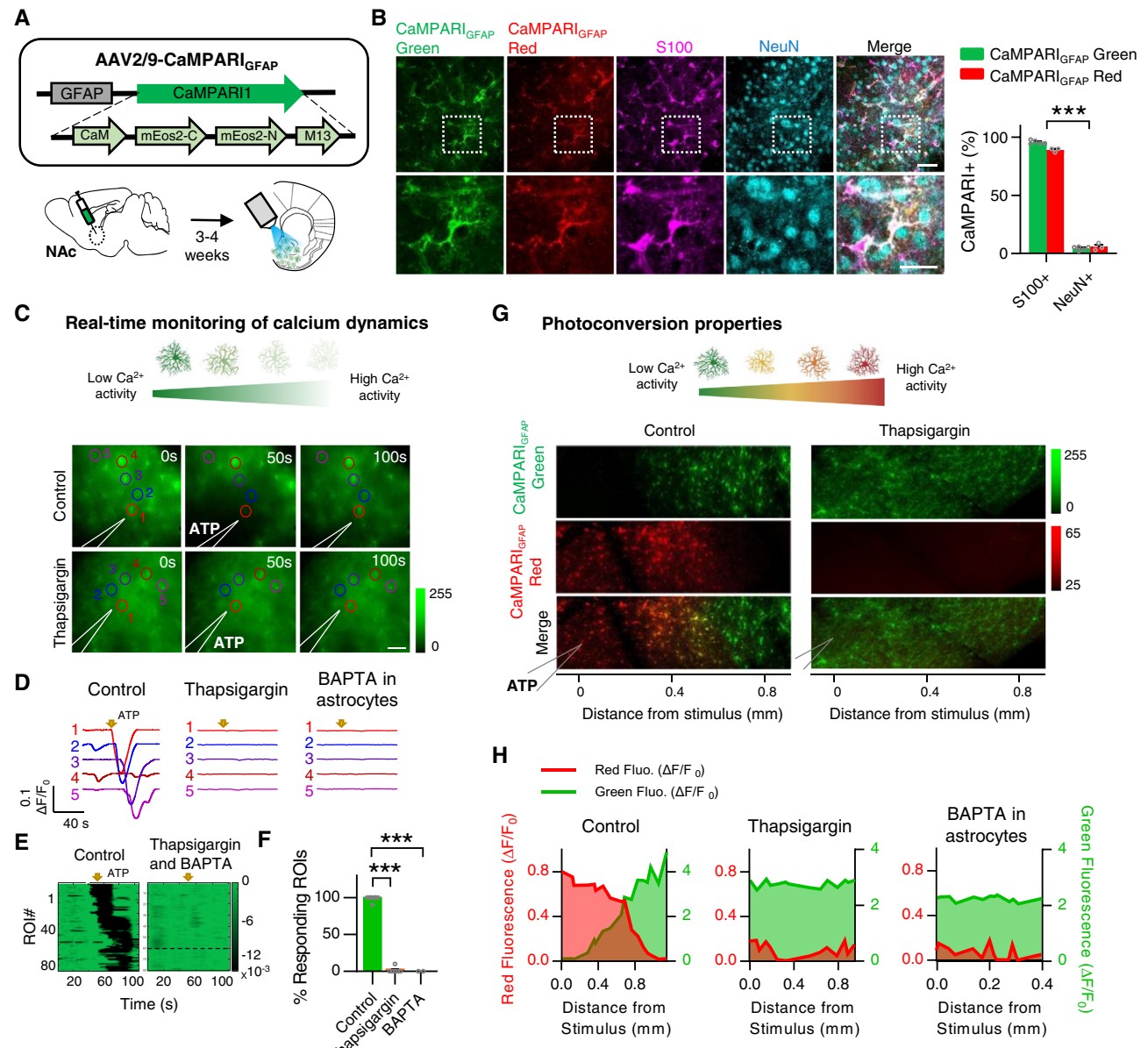

**Fig. 1 | Development of CaMPARI$_{GFAP}$ as a functional tool to study astrocytic networks. A** Top, CaMPARI1 was expressed within an adeno-associated virus (AAV) vector system, under the glial fibrillary acidic protein (GFAP) astroglial promoter. Bottom, CaMPARI$_{GFAP}$ was injected in the nucleus accumbens (NAc) for acute slice imaging. **B** Left, confocal images of CaMPARI$_{GFAP}$ expression in NAc astrocytes. Fluorescence signals show green-to-red photoconversion after illumination with 405 nm light. Magenta and cyan show inmunolabeling with astrocytic (S100) and neuronal (NeuN) markers (right; scale bar = 100 μm) and their inset (bottom; scale bar = 30 μm). Right, quantification showing S100 and NeuN cell populations with percentages of CaMPARI$_{GFAP}$ green- and red-positive cells (8 fields, 2 mice). Two-way ANOVA, \*\*\*$P < 0.001$. **C** Top, scheme of CaMPARI$_{GFAP}$ green fluorescence as a negative calcium indicator, when CaMPARI$_{GFAP}$ binds Ca$^{2+}$ its fluorescence decreases. Bottom, time-lapse of astrocytic activity monitoring CaMPARI$_{GFAP}$ green fluorescence changes in acute slices responding to ATP (20 mM) stimulus in control condition and after perfusion of thapsigargin (1 μM). Scale bar, 50 μm. **D** Astrocytic

Ca$^{2+}$ traces evoked by ATP (20 mM, yellow arrow) in the control condition, in the presence of thapsigargin (1 μM), and after intracellular BAPTA infusion.
**E** Representative heatmaps of ROIs activity, color-coded according to fluorescence change versus time, before and after local ATP application in control (left) and presence of thapsigargin (right; above line) or BAPTA infusion (right; below line).
**F** Quantification of ROIs responding to ATP in control (green; 9 slices, 2 mice), thapsigargin (orange; 6 slices, 2 mice), and BAPTA (purple; 2 slices, 1 mouse) conditions. One-way ANOVA, Holm–Sidak for multiple comparisons, \*\*\*$P < 0.001$.
**G** Top, scheme of CaMPARI$_{GFAP}$ photoconversion color code: green (low Ca$^{2+}$ activity) to red (high Ca$^{2+}$ activity). Bottom, representative CaMPARI$_{GFAP}$ green, red, and merged fluorescence of astrocytes after ATP local stimulation (20 mM) and 10 s 405 nm light pulse in control condition and with thapsigargin (1 μM).
**H** CaMPARI$_{GFAP}$ green and red fluorescence quantification (ΔF/F$_0$) vs distance from ATP stimulus (0 mm refers to ATP-filled pipette). Error bars express SEM. Source data are provided as a Source Data file.

To investigate this, we performed a spatial analysis of the astrocytic response covering the NAc, taking advantage of the photoconversion of CaMPARI$_{GFAP}$ in a Ca$^{2+}$-dependent manner (Fig. 3). To avoid crosstalk between photoconverted CaMPARI$_{GFAP}$ red signals and tdTom from ChrimsonR-expression axons, these experiments were

performed using a different reporter (EYFP) fused to ChR2 (AAV5-CaMKIIa-hChR2(H134R)-EYFP). Firstly, we developed an analysis method based on the alignment of every NAc slice to a reference mask followed by the regular division of the area in 50-μm squares (Supplementary Fig. S6, see "Methods"). With this method, called "partition

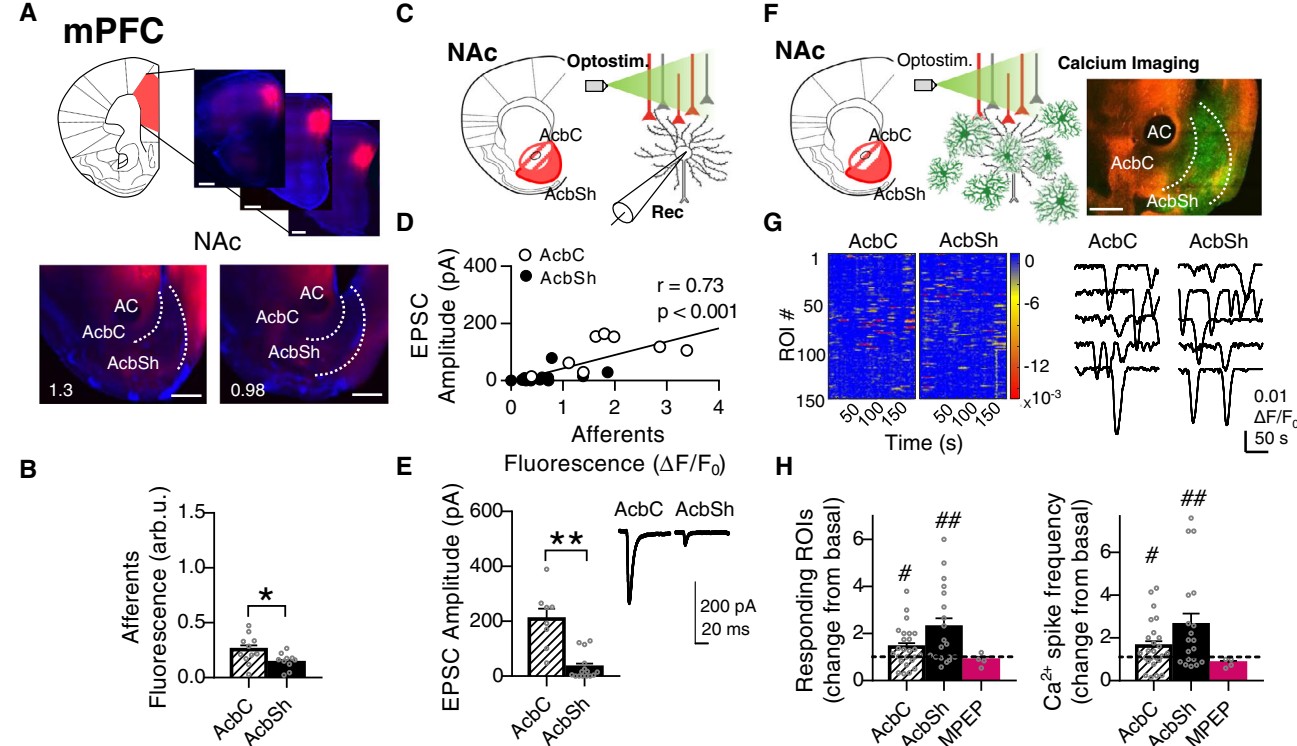

**Fig. 2 | Pathway-specific nucleus accumbens (NAc) astrocyte activity in response to medial prefrontal cortex (mPFC) afferents. A** Scheme and representative slices showing opsin expression (ChrimsonR-tdTom) after AAV injection into the mPFC (top; scale bar = 1 mm) and their tdTom-expressing afferents to the NAc (bottom; scale bar = 500 μm). **B** Quantification of mPFC afferents fluorescence (arb.u.) in accumbens core (AcbC; slashed bar) and accumbens shell (AcbSh; solid bar) (11 infections, 6 mice). Two-tailed unpaired $t$ test, *$P = 0.02$. **C** Scheme of NAc neuron's electrophysiological recordings in response to mPFC afferent's optostimulation. **D** Positive correlation between NAc neuron's response (pA) and mPFC afferents' density ($\Delta F/F_0$ in the area of register) (20 cells, 3 mice). **E** EPSC amplitude (pA) quantification (left) from AcbC (slashed bar; 8 cells, 4 mice) and AcbSh neurons (solid bar; 15 cells, 4 mice) and representative EPSCs traces (right) showing the synaptic neuronal strength triggered by mPFC in each subregion. Two-tailed unpaired $t$ test Welch-corrected; **$P = 0.02$. **F** Scheme of astrocytic $Ca^{2+}$ dynamics,

monitored by real-time imaging of CaMPARI$_{GFAP}$ green fluorescence, in response to mPFC afferent's optostimulation. Representative NAc slice (right) showing ChrimsonR-tdTom mPFC afferents (red) and CaMPARI$_{GFAP}$ astrocytes (green). Scale bar = 500 μm. **G** Heatmaps showing $Ca^{2+}$ activity vs time (left; 150 ROIs, 3 slices) and representative $Ca^{2+}$ traces (right), in response to mPFC optostimulation. **H** Proportion of responding ROIs (left) and relative $Ca^{2+}$ spike frequency (right) induced by mPFC afferent's optostimulation, at AcbC (slashed bar; 25 slices, 8 mice) and AcbSh (solid bar; 19 slices, 8 mice). Subsets of those ROIs were registered in the presence of the selective mGluR5 antagonist (MPEP, 50 μM) (magenta bars; AcbC and AcbSh pooled together, 4 slices, 2 mice). One-sample $t$ test, #$P < 0.05$; ##$P < 0.01$; one-way ANOVA, Holm–Sidak test for multiple comparisons, $P > 0.05$. For more statistical detail, see Supplementary Table 1. Error bars express SEM. Source data are provided as a Source Data file.

in regular quadrants" (PRQ), we first obtained the spatial profiles of mPFC afferents to the NAc (Fig. 3A, B). In agreement with our previous characterization (Fig. 2B), fluorescence PRQ analysis detected a higher density of projections in the AcbC ($1.88 \pm 0.03$ arb.u.) compared to the AcbSh ($0.98 \pm 0.06$ arb.u.) ($P < 0.001$; $n = 9$ slices, 6 mice; Fig. 3B), supporting PRQ as a useful method to analyze fluorescence signals in broad areas. Second, we spatially analyzed NAc astrocytic responses to mPFC afferent optostimulation (Fig. 3C, D). To trigger the red fluorescence of CaMPARI$_{GFAP}$, 40 s of 405 nm light was delivered after optogenetic stimulation. Optostimulation of mPFC afferents led to a significant astrocytic photoconversion in both NAc subregions, AcbC ($1.64 \pm 0.16$ change from basal, p = 0.0035) and AcbSh ($1.92 \pm 0.19$ change from basal, $P = 0.0014$) ($n = 18$ slices in 9 pairs, 6 mice; Fig. 3D), showing the same activation profiles as those registered by $Ca^{2+}$ real-time imaging analysis (Fig. 2H). In line with the recorded $Ca^{2+}$ signals, this photoconversion was mediated by mGluR5 receptors, since both regions reduced their activity to basal levels in the presence of MPEP ($0.97 \pm 0.14$ change from basal, $P = 0.8354$) ($n = 8$ slices in 4 pairs, 2 mice; Fig. 3D).

To further ascertain the spatial overlap between the afferents and the activated astrocytic area, we analyzed these two PRQ spatial profiles (mPFC innervation pattern (Fig. 3A) and NAc astrocytic activity pattern (Fig. 3C)), defining the active areas by $k$-mean

clustering (Supplementary Fig. S7, see "Methods"). Using this approach, we identified the activation threshold for each fluorescence signal, generating one mask containing the area with glutamatergic innervation and another containing the NAc area showing astrocytic activation in each individual slice (Fig. 3E). The overlapping region between these two masks ($16.2 \pm 2.94\%$ area of NAc) was significantly smaller when compared to the afferents area ($30.8 \pm 0.75\%$ area of NAc, $P = 0.04$) or to the activated astrocytic regions ($40.9 \pm 6.66\%$ area of NAc, $P = 0.001$) (Fig. 3F), indicating a low rate of spatial overlap between both signals. In addition, we studied the overlapping degree and spatial distributions within the AcbC and AcbSh using a bivariate similarity index[43,44] and found that astrocytes in both subregions were interacting in the same way with the afferents, showing in both cases low overlapping rate (MANOVA, $d = 0$, $P = 0.586$; Fig. 3F). Furthermore, these findings were maintained when local synaptic activity was blocked with tetrodotoxin (TTX; 1 μM) (Supplementary Fig. S8), suggesting that the astrocytic activity reported is not induced by indirect activation of NAc neurons.

Taken together, these results demonstrate that astrocytes from both NAc subregions do not respond in areas under the direct reach of the glutamatergic afferents, showing pathway-specific astrocyte activity triggered by mPFC inputs.

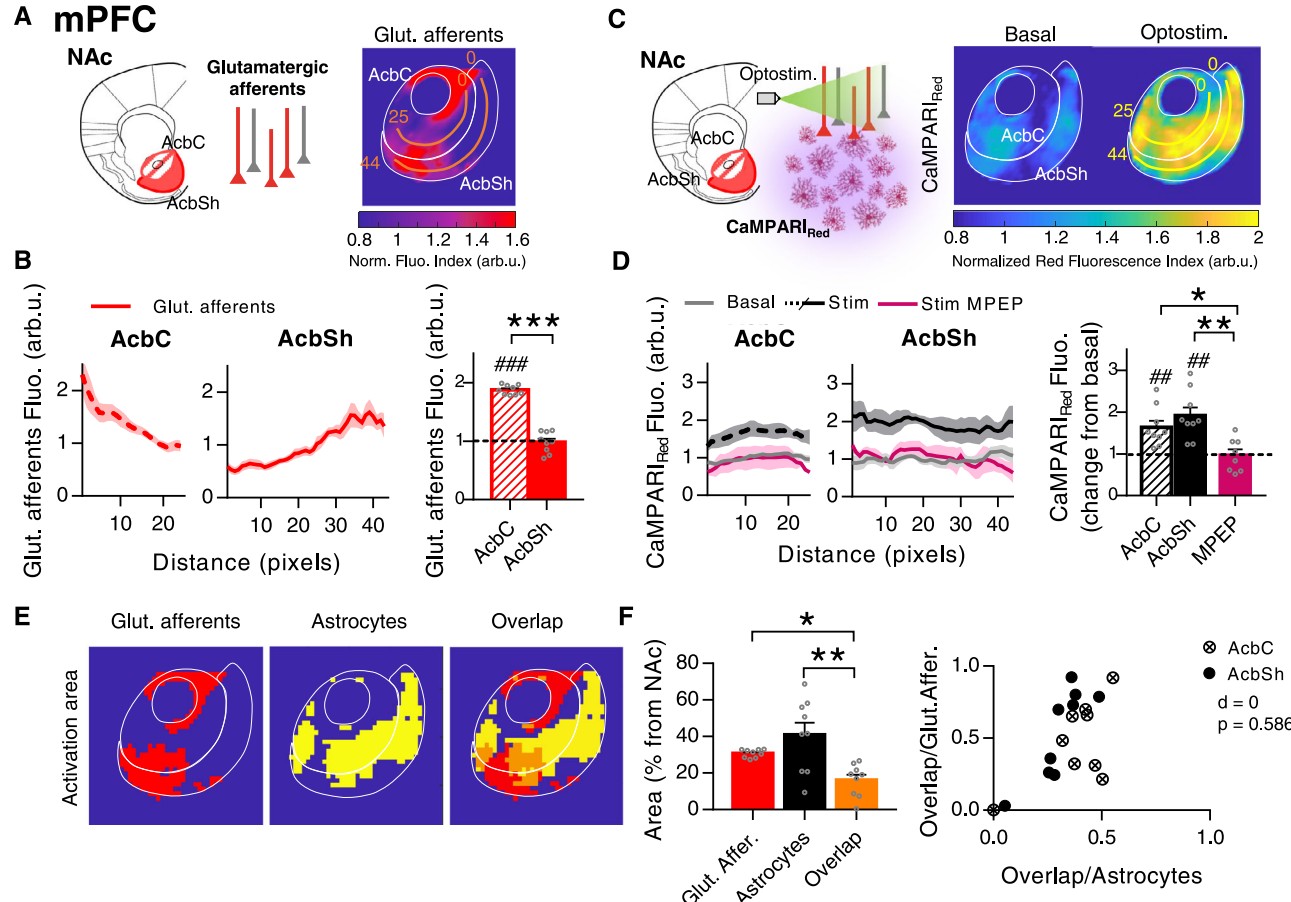

**Fig. 3 | Spatial nucleus accumbens (NAc) astrocytic response to medial pre-frontal cortex (mPFC) afferents. A** Left, scheme of opsin-transfected mPFC afferents in the NAc. Right, average PRQ image showing mPFC glutamatergic innervation pattern. Orange lines starting from pixel 0 in each subregion were used for quantification (pixel = 50 μm²). **B** Left, Glut. afferents fluorescence (arb.u.) vs distance (pixels) quantifying mPFC innervation across orange lines. Right, average glutamatergic afferents spatial fluorescence (arb.u.) in the accumbens core (AcbC; slashed bar) and accumbens shell (AcbSh; solid bar) (9 slices, 6 mice). One-sample *t* test, ###*P* < 0.001; two-tailed unpaired *t* test, ***P* < 0.001. **C** Left, scheme of astro-cytic Ca²⁺ activity, measured by CaMPARI_GFAP red, in response to mPFC afferent's optostimulation. Right, average PRQ image showing astrocytic activation pattern in the NAc in basal and optostimulated conditions. Yellow lines starting from pixel 0 in each subregion were used for quantification (pixel = 50 μm²). **D** Left, CaMPARI_Red fluorescence (arb.u.) vs distance (pixels) quantifying astrocytic activation across

yellow lines. Right, average CaMPARI_Red spatial fluorescence (arb.u.) in optosti-mulated slices in control, at AcbC (slashed bar) and AcbSh (solid bar) (9 pairs basal-stim slices, 6 mice), and in presence of MPEP (magenta bar; AcbC and AcbSh pooled together, 4 pairs basal-stim slices, 2 mice). One-sample *t* test, ##*P* < 0.01; one-way ANOVA, Holm–Sidak test for multiple comparisons, **P* < 0.05, ***P* < 0.01. For more statistic detail, see Supplementary Table 1. **E** Masks of mPFC glutamatergic affer-ents (red) and astrocyte activation area (yellow) defined by a k-mean clustering. In orange, the overlap area between the two. **F** Left, area (% from NAc) quantification of the spatial overlap (orange bar) between mPFC afferents (red bar; **P* = 0.04) and active astrocytes (black bar; ***P* = 0.001) (9 slices, 6 mice). One-way ANOVA, Holm–Sidak test for multiple comparisons. Right, bivariance index showing overlap distributions of AcbC and AcbSh; indicates astrocytes interaction with mPFC afferents. Note that there is no difference between subregions. MANOVA, *d* = 0, *P* = 0.586. Error bars express SEM. Source data are provided as a Source Data file.

## Optostimulation of the Amyg reveals pathway-specific activity patterns in NAc astrocytes

The NAc receives a broadly distributed glutamatergic projection ori-ginating in the Amyg that promotes reward-seeking behaviors[45]. To characterize the NAc astrocyte responses to Amyg inputs, the same approach as above was used. In agreement with previous studies[41,42], histological fluorescence characterization of glutamatergic afferent patterns confirmed that both the AcbC and AcbSh were similarly innervated by Amyg (AcbC 0.37 ± 0.06 arb.u., AcbSh 0.33 ± 0.06 arb.u) (*P* = 0.63, *n* = 13, 7 mice; Fig. 4A, B). PRQ analysis of the spatial dis-tribution of the afferents evidenced that although innervation was similar within subregions, the AcbC (1.34 ± 0.08 arb.u.) was more innervated than the AcbSh (1.14 ± 0.05 arb.u.) (*P* = 0.04, *n* = 9 slices, 6 mice; Fig. 5B). When studying neuronal responses to the optostimu-lation of those axons (Fig. 4C–E), we registered a positive correlation (Pearson *r* = 0.57, *P* = 0.03) between the amplitude of the EPSCs and the density of afferent projections in the recording area (*n* = 15 cells, 2 mice; Fig. 4D). Accordingly, the EPSC amplitudes in each subregion

were similar (AcbC neurons 99.6 ± 35.2 pA, *n* = 11 cells, 3 mice; AcbSh neurons 79.4 ± 15.4 pA, *n* = 25 cells, 4 mice; *P* = 0.54, Fig. 4E).

Real-time CaMPARI_GFAP green fluorescence signal in response to optostimulation of those afferents (Fig. 4F–H) showed that astro-cytes from the AcbSh (2.04 ± 0.38 change from basal, *P* = 0.015) dis-played higher activity compared to those from the AcbC (1.45 ± 0.21 change from basal, *P* = 0.049) (1550 ROIs; *n* = 17 slices, 7 mice; Fig. 4H, left). These responses were mediated in AcbC and AcbSh by mGluR5, as when it was blocked by bath-application of MPEP (50 μM; 1.26 ± 0.28 change from basal, *P* = 0.37; *n* = 5 slices, 2 mice; Fig. 4H, left). Furthermore, CaMPARI_GFAP red measured by spatial PRQ ana-lysis (Fig. 5C, D) showed the same profile, with stronger astrocytic activation in the AcbSh (1.78 ± 0.21 change from basal, *P* = 0.005) compared to the AcbC (1.49 ± 0.18 change from basal, *P* = 0.026) (*n* = 18 slices in 9 pairs, 6 mice) (Fig. 5D). In line with Ca²⁺ imaging data, bath-application of the mGluR5 antagonist reduced those responses to basal levels (0.93 ± 0.10 change from basal, *P* = 0.5) (*n* = 8 slices in 4 pairs; Fig. 5D).

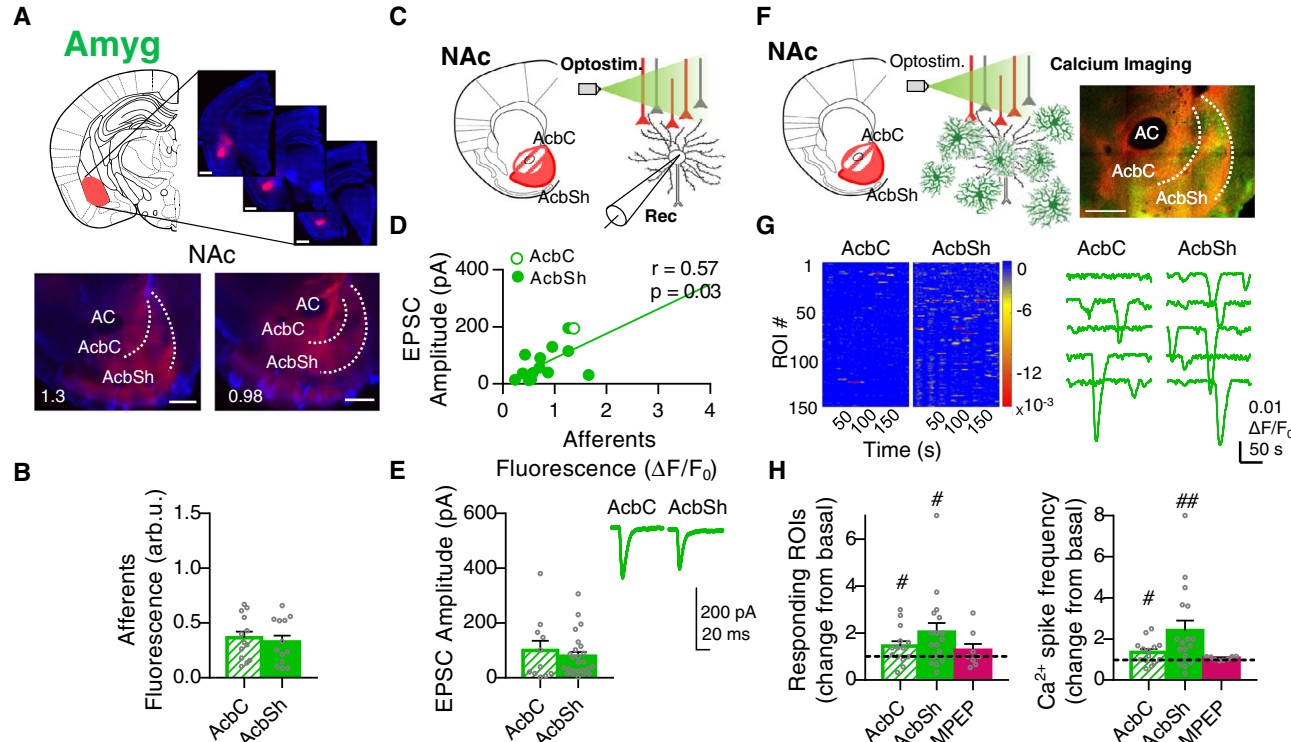

**Fig. 4 | Pathway-specific nucleus accumbens (NAc) astrocyte activity in response to basolateral amygdala (Amyg) afferents. A** Scheme and representative slices showing opsin expression (ChrimsonR-tdTom) after AAV injection into the Amyg (top; scale bar = 1 mm) and their tdTom-expressing afferents to the NAc (bottom; scale bar = 500 μm) (13 infections, 7 mice). **B** Quantification of Amyg afferents fluorescence (arb.u.) in accumbens core (AcbC; slashed bar) and accumbens shell (AcbSh; solid bar) (13 infections, 7 mice). Two-tailed unpaired $t$ test Welch-corrected, $P = 0.63$. **C** Scheme of NAc neuron's electrophysiological recordings in response to Amyg afferent's optostimulation. **D** Positive correlation between NAc neuron's response (pA) and Amyg afferents' density ($\Delta F/F_0$ in the area of register) (15 cells, 2 mice). **E** EPSC amplitude (pA) quantification (left) from AcbC (slashed bar; 11 cells, 3 mice) and AcbSh neurons (solid bar; 25 cells, 2 mice), and representative EPSCs traces (right) showing the synaptic neuronal strength triggered by Amyg in each subregion. Two-tailed unpaired $t$ test, $P = 0.54$. **F** Scheme of astrocytic $Ca^{2+}$ dynamics

monitored by real-time imaging of $CaMPARI_{GFAP}$ green fluorescence, in response to Amyg afferent's optostimulation. Representative NAc slice (right) showing ChrimsonR-tdTom Amyg afferents (red) and $CaMPARI_{GFAP}$ astrocytes (green). Scale bar = 500 μm. **G** Heatmaps (left; 150 ROIs, 3 slices) showing $Ca^{2+}$ activity vs time and representative $Ca^{2+}$ traces (right), in response to Amyg optostimulation. **H** Relative quantification of ROIs response (left) and $Ca^{2+}$ spike frequency (right) induced by Amyg afferent's optostimulation in control condition, at AcbC (slashed bar; 14 slices, 7 mice) and AcbSh (solid bar; 17 slices, 7 mice). Subsets of those ROIs were registered in the presence of the selective mGluR5 antagonist (MPEP, 50 μM; magenta bars; AcbC and AcbSh pooled together, 5 slices, 2 mice). One-sample $t$ test, #$P < 0.05$; ##$P < 0.01$; one-way ANOVA, Holm–Sidak test for multiple comparisons, $P > 0.05$. For more statistical detail, see Supplementary Table 1. Error bars express SEM. Source data are provided as a Source Data file.

Afterward, we compared the area of overlap by calculating masks containing the spatial innervation pattern of Amyg afferents and the activation pattern of NAc astrocytes[46] (Fig. 5E, F). We found that the areas with increased astrocytic activity in response to Amyg optostimulation ($34.5 \pm 5.34\%$ area from NAc, $P = 0.003$ vs overlap) did not spatially match the areas containing more density of glutamatergic projections ($36.8 \pm 1.32\%$ area from NAc, $P < 0.001$ vs overlap), as shown by the low rate of overlap between them ($17.5 \pm 2.05\%$ area from NAc) ($n = 9$ slices, 6 mice; Fig. 5F). Interestingly, astrocytes displayed bigger responses in dorsal regions of the NAc, while axons preferentially innervated ventral areas of the nucleus (Fig. 5E). We did not observe significant differences in this interaction between AcbC and AcbSh astrocytic responses (MANOVA, $d = 0$, $P = 0.482$; Fig. 5F). These results showed that Amyg optostimulation provoked a spatial profile of NAc astrocyte activation different than that of Amyg glutamatergic innervation. Astrocytes in the dorsal areas of the NAc showed a low degree of innervation and higher activity.

### Astrocytes from the NAc show pathway-specific activation motifs in response to vHip glutamatergic afferents

After characterizing astrocytic activity in response to the Amyg and mPFC, we selectively activated another principal glutamatergic input implicated in anxiety-like behavior and social interaction: the

vHip[16,18,47]. Using the same optogenetic approach, the histological evidence (Fig. 6A, B) showed stronger innervation coming from the vHip to the AcbSh ($0.80 \pm 0.15$ arb.u.) with respect to the AcbC ($0.31 \pm 0.06$ arb.u.) ($P = 0.01$; $n = 10$, 6 mice; Fig. 6B). Functional characterization of these afferents (Fig. 6C–E) showed a correlation (Pearson $r = 0.56$, $P = 0.005$; $n = 23$ cells, 5 mice; Fig. 6D), as observed on the other two glutamatergic projection nuclei (mPFC and Amyg). Neurons in the NAc displayed bigger EPSC events in areas with higher density of vHip axons, and the synaptic neuronal strength was substantially stronger in the AcbSh ($303.4 \pm 62.5$ pA; $n = 21$ cells, 6 mice) than in the AcbC ($24.1 \pm 14$ pA; $n = 4$ cells, 3 mice; $P < 0.001$; Fig. 6E). These results confirm that the EPSC amplitude was dependent on the density of glutamatergic fibers, being a common characteristic of the three glutamatergic pathways.

Next, we studied the astrocytic responses to those afferents. Real-time $CaMPARI_{GFAP}$ green fluorescence signal (Fig. 6F–H) showed increased $Ca^{2+}$ spike frequency in the AcbC ($2.73 \pm 0.28$ change from basal, $P < 0.001$) in response to optostimulation with respect to astrocytes from the AcbSh ($2.44 \pm 0.3$ change from basal, $P < 0.001$) (2800 ROIs, $n = 28$ slices, 8 mice; Fig. 6H, right). The astrocytic activity pattern was stronger in the AcbC, in opposition to the innervation profile in which the AcbSh gathered the majority of incoming afferents (Fig. 6B, H). As we found for the two other studied glutamatergic

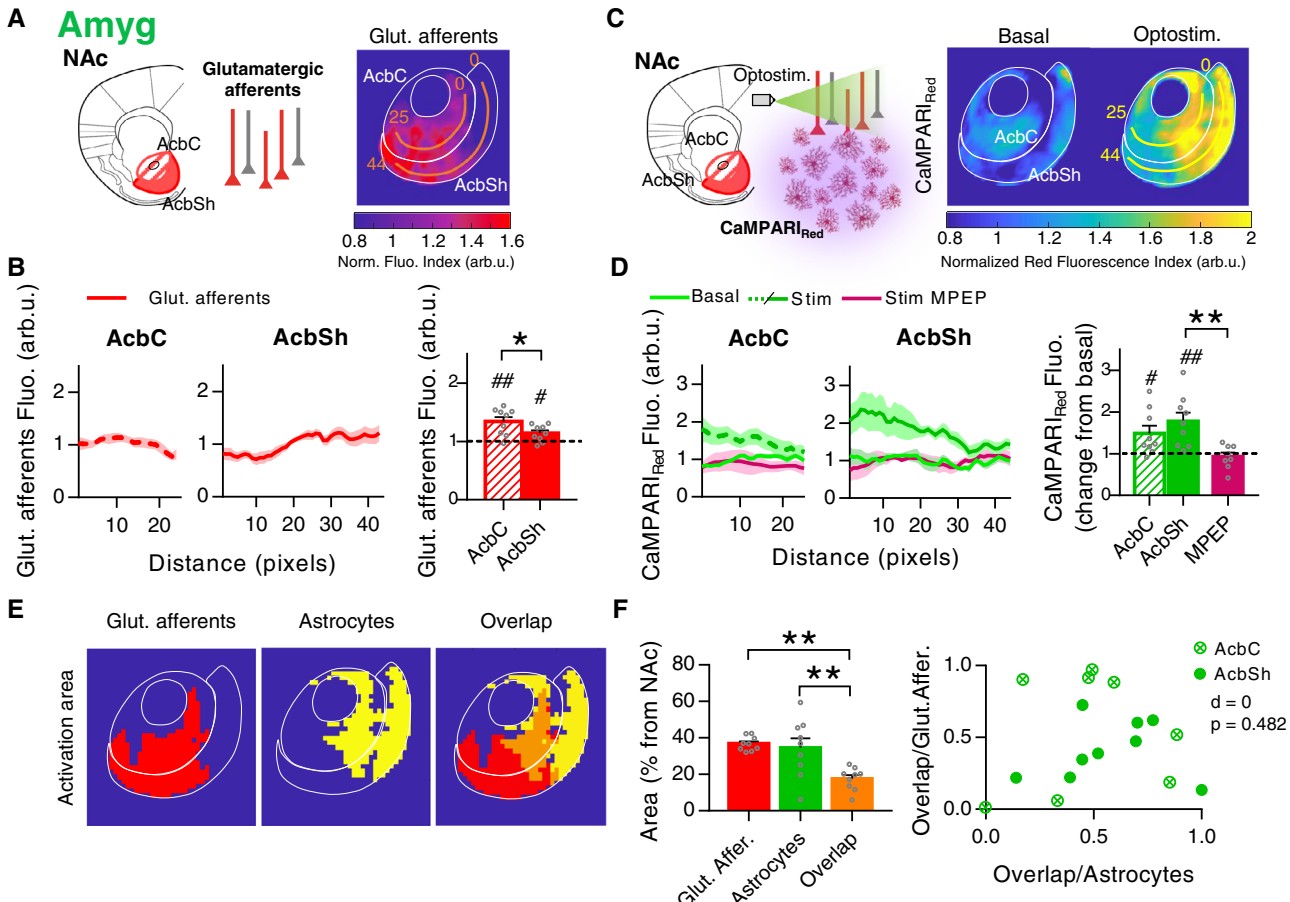

**Fig. 5 | Spatial nucleus accumbens (NAc) astrocytic response to basolateral amygdala (Amyg) afferents. A** Left, scheme of opsin-transfected Amyg afferents in the NAc. Right, average PRQ image showing Amyg glutamatergic innervation pattern. Orange lines starting from pixel 0 in each subregion were used for quantification (pixel = 50 μm²). **B** Left, glutamate afferents fluorescence (arb.u.) vs distance (pixels) quantifying Amyg innervation across orange lines. Right, average glutamatergic afferents spatial fluorescence (arb.u.) in the accumbens core (AcbC; slashed bar) and accumbens shell (AcbSh; solid bar) (9 slices, 6 mice). One-sample $t$ test, $^{\#}P = 0.017$, $^{\#\#}P = 0.0021$; two-tailed unpaired $t$ test, $^{*}P = 0.04$. **C** Left, scheme of astrocytic Ca²⁺ activity, measured by CaMPARI_GFAP red, in response to Amyg afferent's optostimulation. Right, average PRQ image showing astrocytic activation pattern in the NAc in basal and optostimulated conditions. Yellow lines starting from pixel 0 in each subregion were used for quantification (pixel = 50 μm²). **D** Left, CaMPARI_Red fluorescence (arb.u.) vs distance (pixels) quantifying astrocytic activation across yellow lines. Right, average CaMPARI_Red spatial fluorescence (arb.u.) in optostimulated condition vs basal values in control, at AcbC (slashed bar) and AcbSh (solid bar) (9 pairs basal-stim slices, 6 mice), and in presence of MPEP (magenta bar; AcbC and AcbSh pooled together, 4 pairs basal-stim slices, 2 mice). One-sample $t$ test, $^{\#}P = 0.0261$, $^{\#\#}P = 0.0054$; One-way ANOVA, Holm−Sidak test for multiple comparisons, $^{**}P = 0.007$. **E** Masks of Amyg glutamatergic afferents (red) and astrocyte activation area (yellow) defined by a k-mean clustering. In orange, the overlap area between the two. **F** Left, area (% from NAc) quantification of the spatial overlap (orange bar) between Amyg afferents (red bar; $^{**}P = 0.001$) and active astrocytes (green bar; $^{**}P = 0.003$) (9 slices, 6 mice). One-way ANOVA, Holm−Sidak test for multiple comparisons. Right, bivariance index showing overlap distributions of AcbC and AcbSh; indicates astrocytes interaction with Amyg afferents. Note that there is no difference between subregions. MANOVA, $d = 0$, $P = 0.482$. Error bars express SEM. Source data are provided as a Source Data file.

inputs, astrocytic responses triggered by vHip axons were mediated by mGluR5 receptors (MPEP, 50 μM; 0.68 ± 0.17 change from basal, $P = 0.09$; $n = 6$ slices; Fig. 6H, right). These results demonstrated that although NAc astrocytes respond differently depending on the pathway, the glutamate-dependent activity in the nucleus is, in all cases, mediated mainly by mGluR5 receptors.

Analysis of the spatial profile of astrocytic activation using CaMPARI_GFAP red and PRQ (Fig. 7) confirmed strong astrocytic responses in the AcbC (2.46 ± 0.32 change from basal, $P = 0.003$) and AcbSh (2.05 ± 0.20 change from basal, $P = 0.0012$; $n = 16$ slices in 8 pairs, 6 mice), which were blocked in the presence of mGluR5 antagonist (MPEP, 50 μM; 1 ± 0.23 change from basal, $P = 0.99$; $n = 8$ slices in 4 pairs, 2 mice; Fig. 7D). Overall, the spatial location of astrocytes activated by vHip glutamatergic afferents differed from the profiles found in the other glutamatergic pathways (mPFC and Amyg), suggesting that astrocytes in the NAc respond differently to the diverse glutamatergic nuclei showing synapse-specific activity.

Interestingly, in contrast to the results found after stimulation of the mPFC and Amyg, PRQ analysis of spatial overlap (23.6 ± 2.50% area of NAc) between glutamatergic afferents from vHip (34.6 ± 1.51% area of NAc, $P = 0.12$) and astrocytic activation (46.8 ± 6.97% area of NAc, $P = 0.003$) showed colocalization in the AcbSh ($n = 8$ slices, 6 mice; Fig. 7F). This interaction was present in the AcbSh subregion but not in the AcbC (MANOVA, $d = 1$, $P = 0.015$; Fig. 7F).

Finally, we focused on NAc-intrinsic properties that could impact the previously recorded astrocytic calcium responses[9,41,48]. First, we conducted a structural analysis of astrocytic density, followed by analysis of the Ca²⁺ signaling evoked by a group I mGluR agonist (DHPG, 50 μM) (Supplementary Fig. S9). No significant differences were detected, neither in density ($P = 0.61$; Supplementary Fig. S9A, B) nor in spatial distribution ($P = 0.84$; Supplementary Fig. S9C−E) of astrocytes, which showed a homogeneous dispersion across the whole nucleus. Furthermore, the astrocyte response to DHPG bath-application showed no differences across subregions ($P = 0.64$;

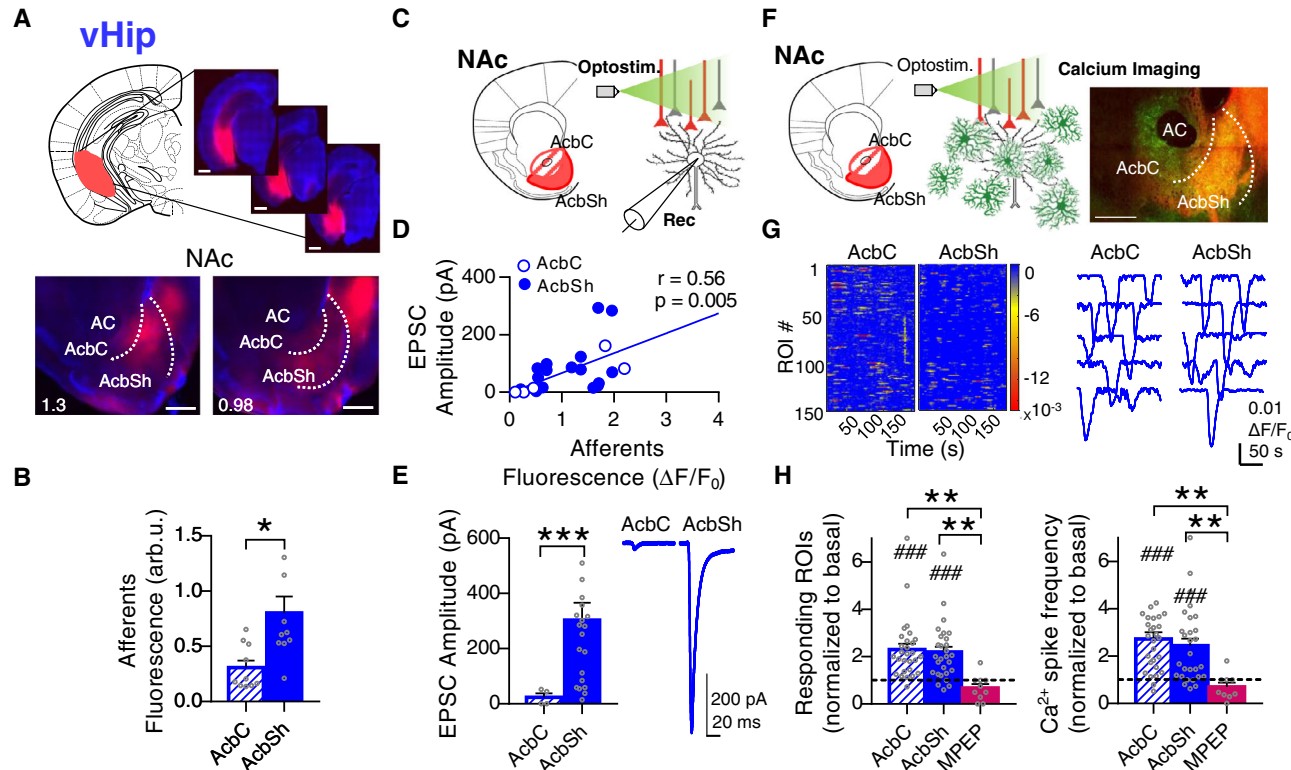

**Fig. 6 | Pathway-specific nucleus accumbens (NAc) astrocyte activity in response to ventral hippocampus (vHip) afferents. A** Scheme and representative slices showing opsin expression after AAV injection into the vHip (top; scale bar = 1 mm) and their tdTom-expressing afferents to the NAc (bottom; scale bar = 500 μm). **B** Quantification of vHip afferents fluorescence (arb.u.) in accumbens core (AcbC; slashed bar) and accumbens shell (AcbSh; solid bar) (10 infections, 6 mice). Two-tailed unpaired $t$ test Welch-corrected, *$P = 0.01$. **C** Scheme of NAc neuron's electrophysiological recordings in response to vHip afferent's optostimulation. **D** Positive correlation between NAc neuron's response (pA) and vHip afferents' density ($\Delta F/F_0$ in the area of register) (23 cells, 5 mice). **E** EPSC amplitude (pA) quantification (left) from AcbC (slashed bar; 4 cells, 3 mice) and AcbSh neurons (solid bar; 21 cells, 6 mice) and representative EPSCs traces (right) showing the synaptic neuronal strength triggered by vHip in each subregion. Two-tailed unpaired $t$ test, ***$P < 0.001$. **F** Scheme of astrocytic $Ca^{2+}$ dynamics, monitored by real-time imaging of CaMPARI$_{GFAP}$ green fluorescence, in response to vHip afferent's optostimulation. Representative NAc slice (right) showing ChrimsonR-tdTom vHip afferents (red) and CaMPARI$_{GFAP}$ astrocytes (green). Scale bar = 500 μm. **G** Heatmaps (left; 150 ROIs, 3 slices) showing $Ca^{2+}$ activity vs time and representative $Ca^{2+}$ traces (right), in response to vHip optostimulation. **H** Quantification (relative to basal) of ROIs response (left) and $Ca^{2+}$ spike frequency (right) in control condition, at AcbC (slashed bar; 28 slices, 8 mice) and AcbSh (solid bar; 28 slices, 8 mice). Subsets of those ROIs were registered in presence of the selective mGluR5 antagonist (MPEP, 50 μM) (magenta bars; AcbC and AcbSh pooled together, 6 slices, 2 mice). One-sample $t$ test, ###$P < 0.001$; one-way ANOVA, Holm−Sidak test for multiple comparisons, **$P < 0.01$; ***$P < 0.001$. For more statistical detail, see Supplementary Table 1. Error bars express SEM. Source data are provided as a Source Data file.

Supplementary Fig. S9F, H), dismissing the existence of specific areas in the nucleus more sensitive to mGluR activation.

Overall, these results show distinct calcium activity of NAc astrocytes in response to vHip afferents, showing an AcbSh interaction between afferents and astrocytic response not present in the other glutamatergic pathways.

### Pathway-specific neuron−astrocyte circuitries within the nucleus accumbens

Next, we explored the differences in astrocytic response between the three glutamatergic inputs, comparing the strength of astrocytic activation among pathways and analyzing the anatomical location of each pattern of astrocytic activity (Fig. 8). As shown in Fig. 8A, B and Supplementary Fig. S5, vHip afferents let to significantly higher fluorescence intensity compared to mPFC (i.e., higher astrocytes $Ca^{2+}$ activity) ($P = 0.024$; Fig. 8B) or Amyg ($P = 0.015$; Fig. 8B). This result demonstrates that although we did not detect specific spatial groups of active astrocytes related to each input (Supplementary Fig. S10), there is a different intensity of astrocytic $Ca^{2+}$ response depending on the pathway, showing synapse specificity and supporting the hypothesis that NAc astrocytes could discern the origin of each of the glutamatergic inputs.

Moreover, our results showed partial spatial overlap between high afferent density areas and areas with high astrocytic $Ca^{2+}$ evoked activity (Figs. 3F, 5F, and 7F). To further ascertain these regions, we compared each map (Fig. 8C, D) performing a pixel-by-pixel correlation for the different pairs of nuclei (Fig. 8D). mPFC vs Amyg (Pearson $r = 0.0985$, $P = 0.009$) and Amyg vs vHip (Pearson $r = 0.3158$, $P < 0.001$) showed r values below 0.5 indicating no correlation, whereas mPFC vs vHip (Pearson $r = −0.2866$, $P < 0.001$) showed a negative correlation. These results show that interaction between astrocytes and high-density afferents are spatially segregated, revealing different neuron-astrocytes circuitries that could point towards the existence of hot-spots in the NAc for each glutamatergic input.

Furthermore, we studied whether the NAc astrocytes exhibit these pathway-specific features in response to other neurotransmitters rather than the glutamatergic system, so we focused on the ventral tegmental area (VTA) inputs to the NAc (Supplementary Figs. S11 and S12). The VTA contains different neurons that co-release dopamine and glutamate, but it has been classically described as a predominantly dopaminergic nucleus[49]. Innervation from this nucleus to the NAc was homogeneous across subregions, as shown by the analysis of afferent's histology ($2.85 \pm 0.53$ and $3.52 \pm 0.49$ arb.u. in the AcbC and AcbSh respectively, $P = 0.38$; Supplementary Fig. S11C). Afterward, we

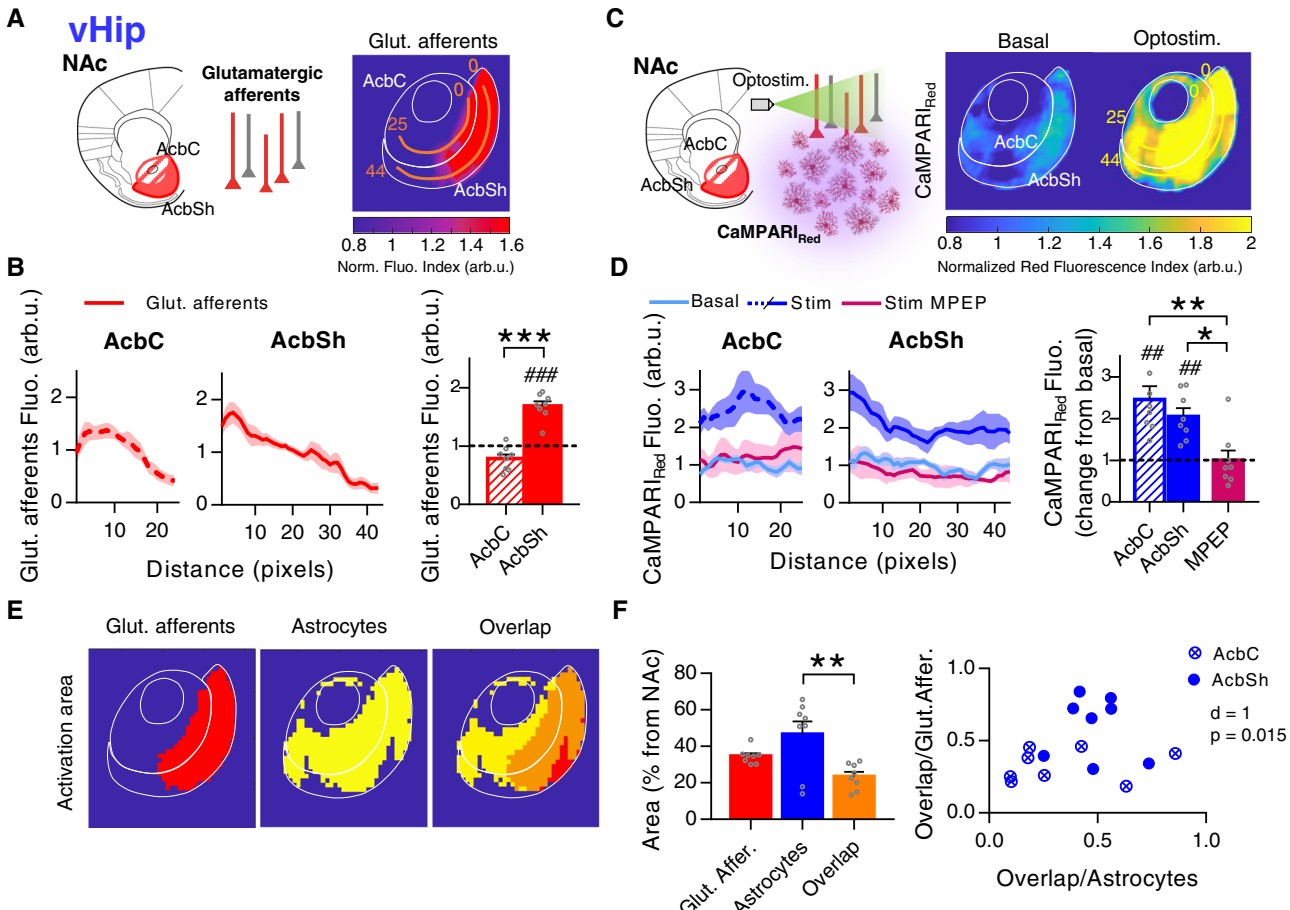

**Fig. 7 | Spatial nucleus accumbens (NAc) astrocytic response to ventral hippocampus (vHip) afferents.** **A** Left, scheme of opsin-transfected vHip afferents in the NAc. Right, average PRQ image showing vHip glutamatergic innervation pattern. Orange lines starting from pixel 0 in each subregion were used for quantification (pixel = 50 μm²). **B** Left, glutamate afferents fluorescence (arb.u.) vs distance (pixels) quantifying vHip innervation across orange lines. Right, average glutamate afferents spatial fluorescence (arb.u.) in the accumbens core (AcbC; slashed bar) and accumbens shell (AcbSh; solid bar) (8 slices, 6 mice). One-sample $t$ test, $^{###}P < 0.001$; two-tailed unpaired $t$ test, $^{***}P < 0.001$. **C** Left, scheme of astrocytic Ca²⁺ activity, measured by CaMPARI$_{GFAP}$ red, in response to vHip afferent's optostimulation. Right, average PRQ image showing astrocytic activation pattern in the NAc in basal and optostimulated conditions. Yellow lines starting from pixel 0 in each subregion were used for quantification (pixel = 50 μm²). **D** Left, CaMPARI$_{Red}$ fluorescence (arb.u.) vs distance (pixels) quantifying astrocytic activation across yellow lines. Right, relative average CaMPARI$_{Red}$ spatial fluorescence (arb.u.) in

optostimulated condition in control, at AcbC (slashed bar) and AcbSh (solid bar) (8 pairs basal-stim slices, 6 mice), and in presence of MPEP (magenta bar; AcbC and AcbSh pooled together, 4 pairs basal-stim slices, 2 mice). One-sample $t$ test, $^{##}P < 0.001$; one-way ANOVA, Holm–Sidak test for multiple comparisons, $^{*}P = 0.02$; $^{**}P = 0.002$. **E** Masks of vHip glutamatergic afferents (red) and astrocyte activation area (yellow) defined by a k-mean clustering. In orange, the overlap area between the two. **F** Left, area (% from NAc) quantification of the spatial overlap (orange bar) between vHip afferents (red bar; $P = 0.12$) and active astrocytes (blue bar; $^{**}P = 0.003$) (8 slices, 6 mice). One-way ANOVA, Holm–Sidak test for multiple comparisons. Right, bivariance index showing overlap distributions of AcbC and AcbSh. Note that AcbSh overlap area is embedded within the afferent's area (Overlap/Glut. Affer. close to 1), indicating a direct interaction between vHip axons and astrocytes not present in AcbC. MANOVA test. Error bars express SEM. Source data are provided as a Source Data file.

registered the astrocytic Ca²⁺ dynamics in response to optostimulation of VTA afferents both in control conditions and after bath perfusion of dopamine receptor antagonists (10 μM haloperidol and 10 μM SCH 23390; Supplementary Fig. S11E). In agreement with previous studies which showed that NAc astrocytes respond to dopamine released from the VTA[50], the optostimulation of these axons increased the frequency of Ca²⁺ spikes in both NAc subregions AcbC (2.06 ± 0.2, change from basal; p = 0.001) and AcbSh (2.04 ± 0.23, change from basal; $P = 0.003$) ($n = 8$ slices, 5 mice), which was blocked in presence of dopamine receptor antagonists (0.99 ± 0.1, change from basal; $P = 0.89$; $n = 9$ slices, 2 mice) (Supplementary Fig. S11E), confirming that this neuron–astrocyte communication was mediated by dopamine. Using CaMPARI$_{GFAP}$ photoconversion and PRQ spatial analysis, we observed increased calcium activity in wide areas of the NAc ($n = 6$ slices in 3 pairs, 3 mice; Supplementary Fig. S12D) and spatial analysis revealed that NAc astrocytes respond in areas directly innervated by VTA axons

($P = 0.07$, Supplementary Fig. S12E, F). Interestingly, after comparison of % overlap area of the different afferents, we found characteristic intrinsic features depending on the glutamatergic or dopaminergic transmission ($P < 0.001$, Fig. 8F). Overall present results show distinct astrocytic calcium activation in response to principal glutamatergic nuclei and define spatially segregated regions enclosing direct interaction between both circuit elements, which are only found in the glutamatergic system.

### Simultaneous activation of different glutamatergic pathways reveals integrative properties of NAc astrocytes

Finally, we investigated how incoming synaptic signals from various glutamatergic pathways were integrated by NAc astrocytes. The ability to integrate multiple synaptic inputs from different entries is a fundamental property of neurons[25]. Similarly, for individual astrocytes, there is evidence of synaptic input integration through Ca²⁺ spike

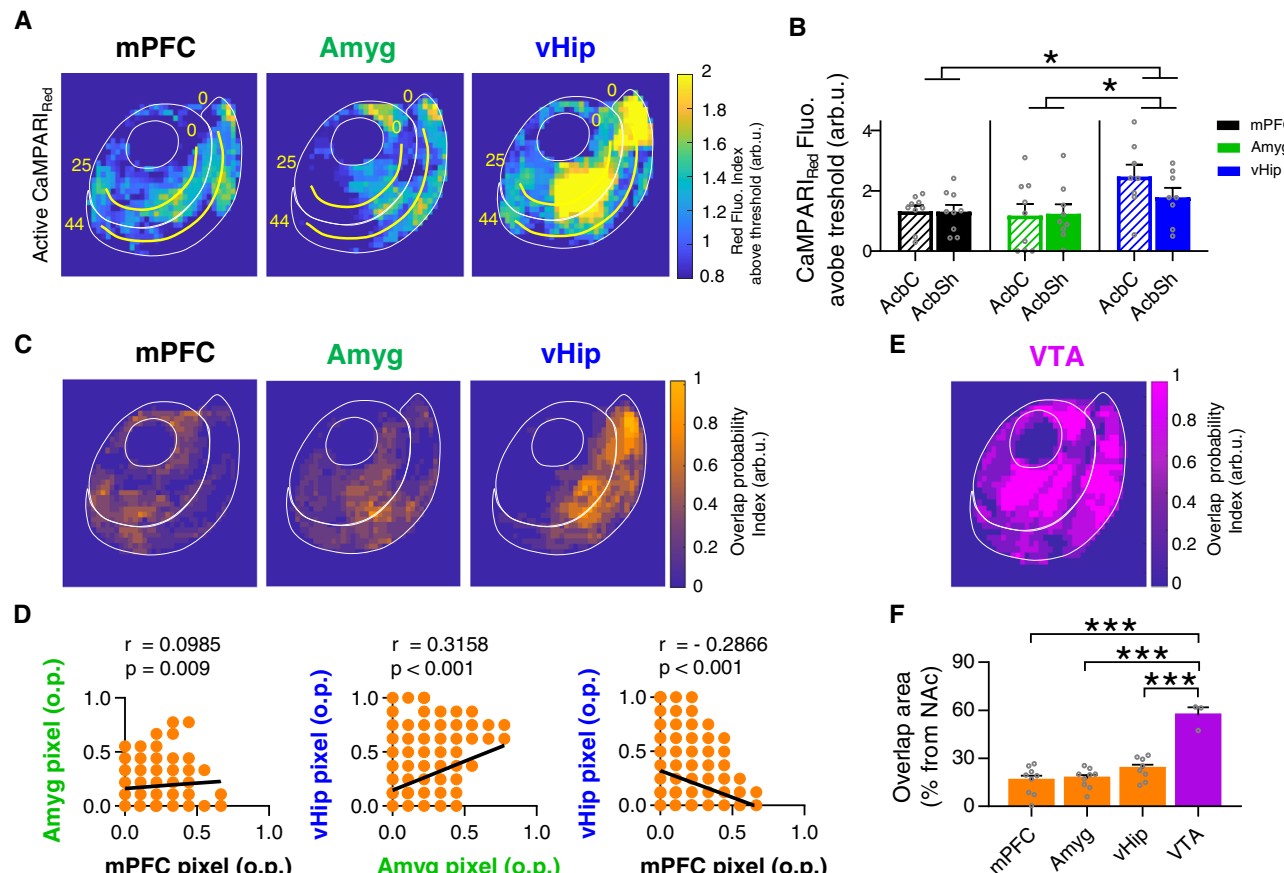

**Fig. 8 | Pathway-specific neuron–astrocyte circuitries in the nucleus accumbens. A** NAc spatial map of astrocytic activity above the activation threshold in response to each glutamatergic input. Yellow lines starting from pixel 0 in each subregion were used for quantification (pixel = 50 μm²). **B** Average CaMPARI$_{Red}$ spatial fluorescence (arb.u.) showing the signal above the activation threshold of the three pathways (mPFC black, 9 slices, 6 mice; Amyg green, 9 slices, 6 mice; vHip blue, 8 slices, 6 mice). Two-way ANOVA, Holm–Sidak test for multiple comparisons; *$P < 0.05$. For more statistical detail, see Supplementary Table 1. **C** NAc spatial maps showing overlap colocalization probability between high-density afferent regions and increased astrocytic activity for each glutamatergic input. **D** Spatial comparisons of the three overlap domains by pixel-to-pixel correlation of different glutamatergic pairs. Pixel value represents the overlap probability (o.p.) for an experimental condition in a specific space (pixel = 50 μm²). Note the low correlation values between conditions, indicating spatial segregation of the overlapping domains. Pearson $r$ correlation, $P$ value (two-tailed). **E** NAc overlap probability map between high-density afferent regions and increased astrocytic activity for ventral tegmental area (VTA) input. **F** Comparisons of the % overlap area among VTA and glutamatergic inputs. One-way ANOVA, Holm–Sidak test for multiple comparisons, ***$P < 0.001$. Error bars express SEM. Source data are provided as a Source Data file.

mediation[25,36]. To test the integrative capacity of NAc astrocytes, we studied Ca²⁺ responses triggered by the combined stimulation of two or three glutamatergic pathways simultaneously. For that, we expressed the ChR2-EYFP opsin in combinations of 2 or 3 glutamatergic nuclei, stimulated all the NAc projecting fibers, and measured the astrocytic response through CaMPARI$_{GFAP}$ photoconversion in the AcbC and AcbSh (Fig. 9).

Co-stimulation of the mPFC and Amyg afferents (Fig. 9A–C) did not trigger significant astrocytic activity neither in the AcbC (1.23 ± 0.26 ΔF/F₀ change from basal, $P = 0.42$) nor in AcbSh (1.18 ± 0.09 ΔF/F₀ change from basal, $P = 0.11$), the latter being significantly lower than in the individual responses ($P = 0.05$; $n = 10$ slices, 3 mice; Fig. 9C). A similar outcome resulted from the co-stimulation of Amyg and vHip axons (Fig. 9D–F), with recorded values close to baseline in the AcbC (1.36 ± 0.32 ΔF/F₀ change from basal, $P = 0.3$) and AcbSh (1.14 ± 0.22 ΔF/F₀ change from basal, $P = 0.55$; $n = 14$ slices, 4 mice; Fig. 9F). Therefore, we found that co-activation of mPFC+Amyg or Amyg+vHip, resulted in weaker responses of astrocytic Ca²⁺ signaling compared to stimulation of single pathways. Conversely, co-stimulation of the mPFC and vHip afferents (Fig. 9G–I) induced strong astrocytic activity in both the AcbC (2.11 ± 0.4 ΔF/F₀ change from basal, $P = 0.039$) and AcbSh (2.24 ± 0.47 ΔF/F₀ change from basal, $P = 0.046$), resulting in a similar Ca²⁺ activity as that triggered by individual pathway optostimulation (AcbC, $P = 0.12$; AcbSh, $P = 0.73$; $n = 12$ slices, 5 mice; Fig. 9I). These different outcomes in astrocytic dynamics depending on the combination of glutamatergic inputs confirm that astrocytes are able to differentially respond according to the pathway from which the inputs are received.

Moreover, when all three pathways were stimulated simultaneously (mPFC + Amyg + vHip) (Fig. 9J–L), we did not find a statistical increase in AcbC astrocytes from basal Ca²⁺ activity (1.03 ± 0.16 ΔF/F₀ change from basal, $P = 0.84$), and remarkably, astrocytes from the AcbSh reduced their activity even below basal values (0.69 ± 0.09 ΔF/F0 change from basal, $P = 0.018$; $n = 12$ slices, 3 mice; Fig. 9J). Strikingly, stimulation of the amygdaloid afferents induced inhibition effects on astrocyte responses in every other pathway with which it is combined (Fig. 9A, D, J). Given these data, it would appear the Amyg has a dominant influence over astrocyte circuitries.

All these differences in Ca²⁺ processing between subregions further suggest the existence of neuron–astrocyte circuitries that work to coordinate responses to glutamatergic afferents.

## Discussion

Numerous studies have demonstrated distinct functional and structural domains within the NAc[9,51–54]. Thus, two primary subregions have been identified to occupy the dorsolateral and ventromedial portions

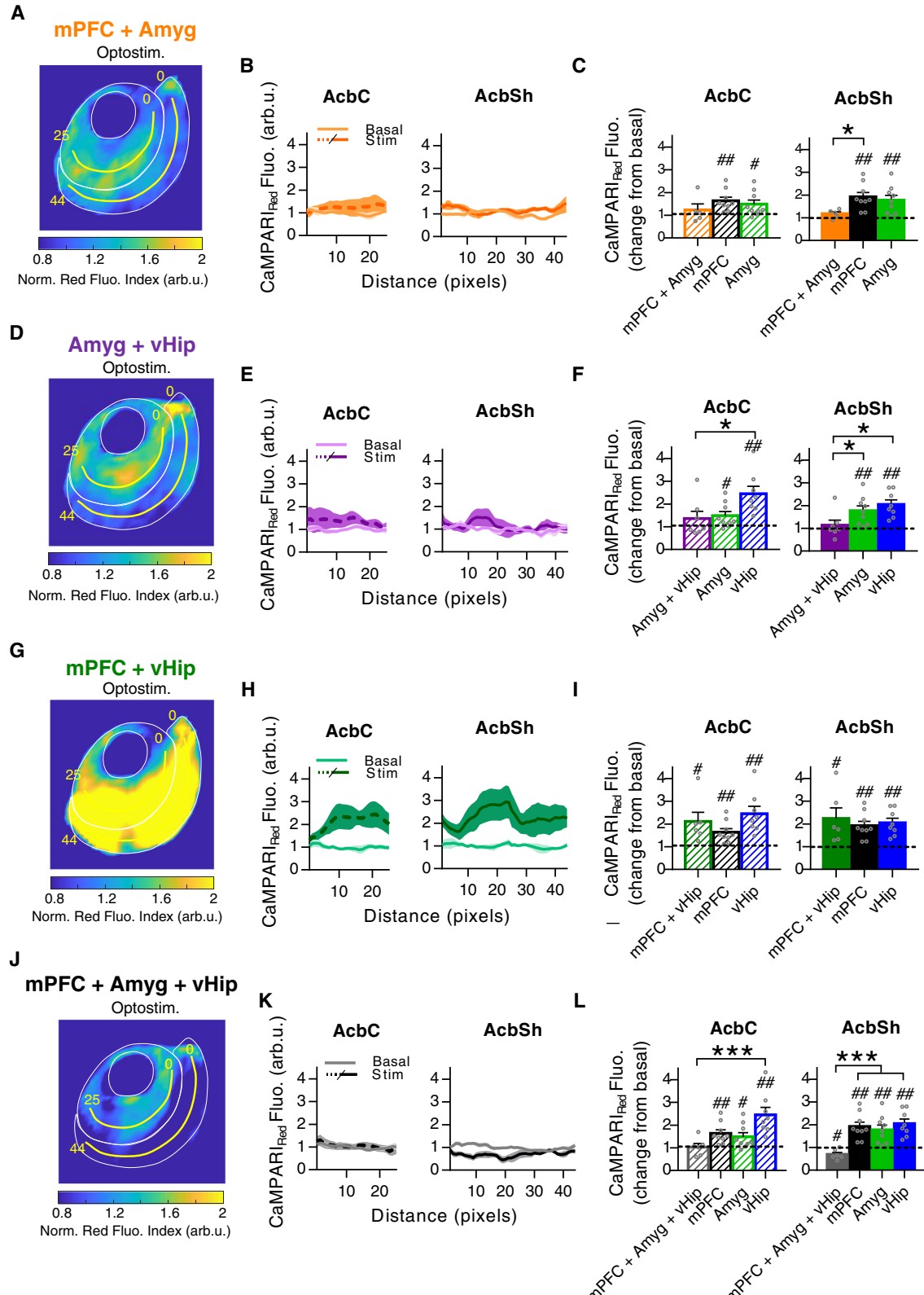

of the NAc, known as the AcbC and AcbSh, respectively. Their different roles are widely attributed to stem from differences in neural circuitry between the regions. Indeed, these roles have traditionally been studied together with structural connectivity patterns determined by axonal tracers[9,55]. Herein, by using a newly developed and specific CaMPARI_GFAP approach, we present data that reveals the existence of synapse-specific neuron–astrocyte circuitries within the NAc.

An imaging technique to simultaneously monitor large numbers of astrocytes is required for higher spatial and temporal studies, which makes CaMPARI_GFAP a powerful tool for unraveling the dynamics of astrocytes and their relationship with the circuits to which they belong. Using this approach, we have therefore been able to capture the astrocytic Ca²⁺ picture occurring in the NAc after glutamatergic stimulation. Preceded by a thorough analysis, we give evidence

**Fig. 9 | Integrative properties of nucleus accumbens (NAc) astrocytes in response to co-stimulation of glutamatergic pathways. A** Average PRQ image showing astrocytic activation pattern in NAc in response to co-stimulation of mPFC and Amyg afferents. **B** CaMPARI$_{Red}$ fluorescence (arb.u.) vs distance (pixels) quantifying astrocytic activation across yellow lines. **C** Average CaMPARI$_{Red}$ spatial fluorescence (arb.u.) in co-stimulated condition vs basal in AcbC and AcbSh (mPFC + Amyg, orange; 5 pairs basal-stim slices, 3 mice), compared to individual mPFC (black) and Amyg (green) response. **D** Average PRQ image showing astrocytic activation pattern in NAc in response to co-stimulation of Amyg and vHip afferents. **E** CaMPARI$_{Red}$ fluorescence (arb.u.) vs distance (pixels) quantifying astrocytic activation across yellow lines. **F** Average CaMPARI$_{Red}$ spatial fluorescence (arb.u.) in co-stimulated condition vs basal in AcbC and AcbSh (Amyg + vHip, purple; 7 pairs basal-stim slices, 4 mice), compared to individual Amyg (green) and vHip (blue) response. **G** Average PRQ image showing astrocytic activation pattern in NAc in response to co-stimulation of mPFC and vHip afferents. **H** CaMPARI$_{Red}$

fluorescence (arb.u.) vs distance (pixels) quantifying astrocytic activation across yellow lines. **I** Average CaMPARI$_{Red}$ spatial fluorescence (arb.u.) in co-stimulated condition vs basal in AcbC and AcbSh (mPFC + vHip, dark green; 6 pairs basal-stim slices, 5 mice), compared to individual mPFC (black) and vHip (blue) response. **J** Average PRQ image showing astrocytic activation pattern in NAc in response to mPFC, Amyg and vHip co-stimulation. **K** CaMPARI$_{Red}$ fluorescence (arb.u.) vs distance (pixels) quantifying astrocytic activation across yellow lines. **L** Average CaMPARI$_{Red}$ spatial fluorescence (arb.u.) in co-stimulated condition vs basal in AcbC and AcbSh (mPFC + Amyg + vHip, gray; 6 pairs basal-stim slices, 3 mice), compared to individual mPFC (black), Amyg (green) and vHip (blue) response. Yellow lines starting from pixel 0 were used for quantification (pixel = 50 μm$^2$). One-sample $t$ test, $^\#P < 0.05$; $^{\#\#}P < 0.01$ and one-way ANOVA, Holm–Sidak test for multiple comparisons, $^*P < 0.05$; $^{***}P < 0.001$. For more statistical detail, see Supplementary Table 1. Error bars express SEM. Source data are provided as a Source Data file.

supporting the existence of unique neuron–astrocyte circuits in each NAc region, which receive and integrate glutamatergic innervation differentially from the mPFC, Amyg, and vHip. Our results reveal a functional heterogeneity between the two NAc subpopulations that leads to differential input integration of the excitatory pathways. Whether these different outcomes of the AcbSh and AcbC Ca$^{2+}$ dynamics are due to particular astrocyte subpopulation heterogeneity, specific properties of the glutamatergic afferents, the total amount of glutamate released into the NAc by the reported pathways, or rather a combination of these, must be further explored. We, therefore, highlight the importance of finding intrinsic features that differ among NAc astrocytes, suggesting physiological heterogeneity between AcbC and AcbSh astrocytes that contribute to condition the overall circuit's response. Furthermore, when considering the critical role of astrocytes for synaptic function, the present results support the idea that different neuron–astrocyte circuitries are associated with discernible functions in NAc; i.e., an astrocyte subpopulation from the AcbSh could be related to the limbic system, and another from the AcbC could be preferentially linked to the motor system.

We found that NAc astrocytes respond to excitatory inputs in a pathway-specific way, since optostimulation of each glutamatergic input results in different profiles of high astrocytic Ca$^{2+}$ activity within the NAc (Fig. 8A, B). Astrocytes did not appear to respond solely in highly innervated regions (Figs. 3E, 5E, and 7E); mPFC inputs triggered high Ca$^{2+}$ activity in astrocytes within both NAc subregions (Figs. 3 and 8), while astrocytes highly responsive to the Amyg seemed to be preferentially concealed within dorsal regions of the NAc (Figs. 5 and 8). Further, vHip activated a broad astrocyte network covering most of the NAc (Figs. 7 and 8). The fact that the astrocytic response is not concealed within the major glutamatergic afferents' regions immediately suggests an underlying complex pattern of network activation. This is a characteristic of glutamatergic pathways that is not present within dopaminergic signaling (Fig. 8F and Supplementary Fig. S12). The existence of these astrocytic networks was revealed by blocking lateral inhibition and forward inhibition using picrotoxin and preserved when neuronal activity was blocked with the Na$^+$ channel blocker TTX, highlighting that defined astrocytic ensembles respond to a specific neuronal input. However, of deep interest would be to further study the implications of activating a profile of astrocytes in less-innervated areas in response to glutamatergic stimuli to enlarge the missing links of the physiological picture. Moreover, we found spatial differences in the way astrocytes interact with high-density afferent areas among the pathways (Fig. 8C–F). Considering the positive correlation between neuronal response and glutamatergic afferents stimulation, it is more likely that these high-density regions constitute the major source of glutamate release in the NAc. Interestingly, the astrocytic response in these areas rich in glutamate is not only restricted, but also segregated in the NAc space for each individual pathway, revealing input-

specific overlap regions. This shows the spatial distribution of different neuron–astrocytes circuitries in the NAc, pointing toward the existence of specific hotspots in the glutamatergic system which are not observed in response to the VTA. Further functional dissection of neuro-glial circuits in other brain regions and in response to other neurotransmitters would be of the outmost interest to bring a deeper understanding and to contrast these hypotheses.

The NAc integrates convergent glutamatergic inputs and this has been traditionally attributed to the membrane properties of the GABAergic MSN (the primary neurons of the NAc), which are considered responsible for mediating this afferent integration[9,22,23,41,42,56,57]. However, we show astrocytes share information and participate in the integration of functional inputs coming from different glutamatergic inputs. The fact that responses coming from simultaneous stimulation of 2 or 3 inputs showed non-linear activity (i.e., that the resulting co-stimulated response is not the sum of the individual responses) indicates that astrocytes may be also able to perform complex computations on the information coming from different sources (see Fig. 9). In a manner that depends on the transient state of the system, dopamine plays a complex role in the gating of afferent input to the NAc[58,59]. According to the data obtained in Fig. 9, there is the possibility that the afferent activity from the Amyg can facilitate the release of mesoaccumbens dopamine efflux in the NAc[59,60]. This co-release of glutamate and dopamine could activate a number of different cellular mechanisms that would lead to an inhibition of Amyg-astrocyte circuitries. These results agree with the reported calcium activity regulation by different synaptic inputs in the hippocampus[25]. While further studies, out of the scope of the present work, are required to elucidate the underlying molecular mechanisms of this phenomenon, it can be hypothesized that it might be due to the interaction of the intracellular signaling pathways stimulated by both synaptic inputs (see refs. 61,62). This lack of linearity shows that the integration property orchestrated by astrocytes in the NAc (e.g., the ability to increase the signal-to-noise ratio (see ref. 63)), could mechanistically explain the divergent physiological and behavioral responses produced by the activation of different glutamatergic inputs to the NAc, revealing the neuron–astrocyte network as a critical center for the integration properties of the NAc[9,22,23,41,42,56,57].

Since the functions of astrocytes appear to be primarily homeostatic[59], and the existence of networks within a system facilitates the ability to self-regulate, it seems plausible that autoregulation is supplied by astrocytic profiles to favor communication under dynamic equilibrium[8,64–66]. This picture of several structurally independent networks entangles points to the existence of a modular system, in which modules connect internally with each other to achieve robustness in the integration and release of signals. Modules may be useful when outputs change over time since reconnections among already existing ones remove the need to form them from scratch[67,68], adding complexity to the network structure with features

such as plasticity, capacity to integrate or adaptability[69]. In addition, modularity may provide a means to finely achieve tuned temporal (such as synchronization) and spatial resolution of signals (discerning among inputs to integrate outputs accordingly). For a holistic understanding of the circuits that mediate diverse behaviors in the NAc, it would probably be necessary to understand the mechanisms by which astrocytes integrate information.

Taken together, the results obtained with the CAMPARI$_{GFAP}$ analysis method support the neuron–astrocyte networks as critical relay participants in the NAc and demonstrate the value of creating comprehensive functional astrocyte-network maps, providing a potential insight for how the NAc integrates information from multiple glutamatergic regions.

# Methods

## Mice

Experiments were performed according to protocols approved by the Institutional Animal Care and Use Committee of the Cajal Institute and by local veterinary authorities (Comunidad de Madrid, PROEX 40/18). C57BL/6J wild-type mice between 1 and 3 months of both sexes were used, and were housed in standard laboratory cages with ad libitum access to food and water, under a 12:12 h dark–light cycle in temperature-controlled rooms (20–22 °C, 45–65% humidity). All animals were obtained from the animal facility of the Cajal Institute (registration number ES280790000184).

## CaMPARI$_{GFAP}$ viral vector

pAAV2/9-hSyn-CaMPARI-WPRE-SV40 (Ca$^{2+}$-modulated photo-activatable ratiometric integrator) plasmid was acquired from Addgene (Item # 100832). The hSyn promoter was substituted by the GFAP promoter short version GFAP104 and cloned into AAV particles by Unitat de Producció de Vectors (UPV) (pAAV2/9-GFAP.CaMPARI-WPRE-SV40; viral titer $1.05 \times 10^{13}$ gc/ml).

## Viral injection

The following constructs were used: AAV5-CaMKIIa-hChR2(H134R)-mcherry, viral titer $5.6 \times 10^{12}$ gc/ml; pAAV9-Syn-ChrimsonR-tdTomato, viral titer $4.1 \times 10^{12}$ gc/ml; AAV5-CaMKIIa-hChR2(H134R)-EYFP, viral titer $6.1 \times 10^{12}$ gc/ml; pAAV2/9-GFAP-CaMPARI-WPRE-SV40; viral titer $1.05 \times 10^{13}$ gc/ml. P15-P20 mice were anesthetized with 2% isoflurane in oxygen and placed in a custom-adapted stereotaxic apparatus. Stereotaxic bilateral injections (300–350 nl; 60 nl/min) were made in NAc (AP: 1.45 mm ML: ± 0.65 mm DV: −4.3 mm) and/or mPFC (AP: 1.7 mm ML: ±0.3 mm DV: −2.3 mm), Amyg (AP: −1.3 mm ML: ±2.95 mm DV: −4.4 mm), vHip (AP: −3.1 mm ML: ±2.95 mm DV: −4.2 mm) and VTA (AP: −3 mm ML: ±0.5 mm DV: −4.3 mm). After injection, the pipette was held in place for 10 min prior to retraction to avoid leakage, then removed and skin sutured. Animals were allowed to recover from anesthesia with the help of heating pads and returned to the cage once they showed regular breathing and locomotion. Three to four weeks after viral injection, location of the virus was confirmed based on CaMPARI$_{GFAP}$, mCherry, tdTom and EYFP expression and in vitro experiments were performed. Hemispheres showing no expression or misplaced location of CaMPARI/opsin infections were discarded. Afterwards, slices containing opsin injection sites and NAc were kept for accurate control of viral transfection (see "Afferent density and opsin transfection").

## Slice preparation

Experiments were performed on slices from mice (1–3 months, both sexes) unless otherwise indicated. Three to four weeks after viral infection, animals were anaesthetized and prior to decapitation, transcardially perfused with ice-cold protective artificial cerebrospinal fluid (NMDG ACSF) containing (in mM): N-methyl-D-glucamine (NMDG) 93, KCl 2.5, NaH$_2$PO$_4$ 1.25, NaHCO$_3$ 30, HEPES 20, glucose 25, thiourea 2, Na-ascorbate 5, Na-pyruvate 3, CaCl$_2$ 0.5, MgCl$_2$ 10. The brain was rapidly removed and placed in ice-cold NMDG ACSF, and slices (300-μm thick) were obtained with a vibratome (Leica Vibratome VT1200S, Germany), maintained at 34 °C in NMDG ACSF for 10 min and transferred for >1 h incubation at room temperature (RT) in standard ACSF containing (in mM): NaCl 119, KCl 2.5, NaH$_2$PO$_4$ 1, MgCl$_2$ 1.2, NaHCO$_3$ 26, CaCl$_2$ 2.5, and glucose 11, and gassed with 95% O$_2$/5% CO$_2$ (pH = 7.3). Slices containing the NAc region were placed into the microscope recording chamber superfused with gassed ACSF. Unless otherwise indicated, ACSF was supplemented with 0.05 mM picrotoxin (GABA$_A$ receptors antagonist) to eliminate the influence of feedforward/lateral inhibition[41,70,71].

## Afferent density and opsin transfection

Injection site and NAc slices were kept and fixed with 4% paraformaldehyde (PFA), 4% sucrose in PBS for 1 h at RT. After fixation, the slices were incubated with DAPI (1.5 μg/ml, Sigma-Aldrich) for 10–15 min and all were mounted in Vectashield antifading mounting medium (Vector Laboratories, Burlingame, CA). Fluorescence images were acquired with a 10x objective in a Leica AF 6500–7000 microscope using Leica LAS AF software. Image analysis was carried out with Image J software (public domain software developed at the US National Institutes of Health [NIH]). Regions of interest (ROIs) were delimited manually in each slice, and average fluorescence inside the regions was expressed as the change ($\Delta F/F_0$) between the averaged ROI value (F) and a background ROI (F$_0$) set in a reference point of the tissue without reporter's fluorescent signal. To measure AAV-transfection at the injection site (Supplementary Fig. S11A–C and S13A–C), same number of AP slices (300-μm thick) were analyzed in each nuclei: mPFC (AP: 2.1 mm to 1.6 mm), Amyg (AP: −1.2 mm to 2.4 mm), vHip (AP: −2.8 mm to 4 mm) and VTA (−3 mm to −3.9 mm). The final transfection degree was calculated as the average fluorescence ($\Delta F/F_0$) of the set of slices. To measure afferent's density in the NAc, ROIs were delimited manually for AcbC and AcbSh and fluorescence intensity ($\Delta F/F_0$) was calculated for each subregion, being F$_0$ the background ROI located in a region outside the NAc with no afferent's innervation. Final afferents fluorescence (arb.u.) was expressed as the individual AcbC and AcbSh ($\Delta F/F_0$) values relative to the transfection degree ($\Delta F/F_0$) measured at their respective injection site.

## Electrophysiology

Recordings from NAc neurons were made using the whole-cell patch-clamp technique. Cells were visualized with an Olympus BX50WI microscope (Olympus Optical, Tokyo, Japan) under a ×40 water immersion objective. Patch electrodes had resistances of 4–10 MΩ when filled with an internal solution that contained (in mM): potassium gluconate 135, KCl 10, HEPES 10, MgCl$_2$ 1, ATP-Na$_2$ 2, titrated with KOH to pH 7.3. Recordings were obtained with PC-ONE amplifiers (Dagan) and pClamp software (Molecular Devices). NAc neurons were voltage clamped at −75 mV, unless otherwise noted. Electrophysiological properties were monitored before and at the end of the experiments. Series and input resistances were monitored throughout the experiment using a −1 mV pulse. Recordings were considered stable when the series and input resistances and stimulus artifact duration did not change >20%. Cells that did not meet these criteria were discarded. To assess neuronal excitability (Supplementary Fig. S2B, C), cells were recorded 5 min before and 5 min after 405 nm light delivery in the presence of picrotoxin (50 μM). Spontaneous currents (sEPSCs) analysis was performed using pClamp software (Molecular Devices).

## Optogenetic stimulation

To determine synaptic strength of glutamatergic afferents, optically evoked excitatory postsynaptic currents (EPSCs) were obtained every 5 s with light pulses (0.1–1 W/cm$^2$) using CoolLED's pE-300white system

through the microscope objective for full-field optostimulation. To enable evoked EPSCs response comparisons, light stimulation strength and pulse duration remained constant for all the recordings (70% LED intensity, 1 ms). Two different wavelengths were used, 470 nm and 565 nm, to stimulate afferents expressing ChR2 or ChrimsonR opsins, respectively. To avoid differences derived from stimulation intensity, light stimulation parameters remained constant for all these recordings (70% intensity, 1 ms). Dose–response curves (Supplementary Fig. S4B) were calculated measuring evoked EPSC amplitudes at different light intensities (10, 30, 50, 70, and 90% LED intensity).

To assess astrocytic response, opsin-transfected afferents were full-field activated at the NAc by an optostimulation protocol (10 pulses 50 ms at 4 Hz−four times, 5 s interval) which remained constant for all experiments to allow for astrocytic Ca²⁺ activity comparison between pathways. Specific parameters of the stimulation intensity for ChrimsonR or ChR2, used respectively to study astrocytic calcium real-time imaging or photoconversion, are specified in the following sections.

## Ca²⁺ imaging

NAc slices (300 μm thick) with astrocytes expressing CaMPARI$_{GFAP}$ and glutamatergic fibers expressing Chrimson-tdTom were used and Ca²⁺ levels were monitored by fluorescence microscopy using real-time imaging of CaMPARI$_{GFAP}$ green fluorescence properties. Astrocytes were imaged with a CCD camera (Hamamatsu C474-95) attached to an Olympus BX50WI microscope (Olympus Optical, Tokyo, Japan), coupled with a ×10 water immersion objective and superfused with gassed ACSF in presence of picrotoxin (0.05 mM). After 5 min to allow picrotoxin perfusion, cells were illuminated through the microscope objective at 300 ms exposure with a LED at 470 nm using CoolLED's pE-300white system, and images were acquired every 1 s at RT. The CoolLED and CCD camera were controlled and synchronized by Nis-Elements Advance Research software (Nikon Instruments Europe B.V.). After 3 min of basal-activity recording, ChrimsonR optostimulation protocol was delivered at a fixed intensity (1–5 mW/cm²) by an external LED (Thorlabs M590F3, λ = 590 nm), followed by 3 min recordings of the astrocytic responses. For a subset of experiments, recordings were performed in the presence of 50 μM MPEP (selective antagonist of mGluR5) or 10 μM haloperidol and 10 μM SCH 23390 (dopamine receptors antagonists). Furthermore, to test specifically CaMPARI$_{GFAP}$ properties as a calcium indicator, astrocytes were activated by a local pulse of 20 mM ATP in the control condition and after extracellular perfusion of 1 μM thapsigargin (endoplasmic reticulum Ca²⁺-ATPase inhibitor), or intracellular loading of 20 mM BAPTA (calcium chelator) in the astrocytic network through a path pipette.

Preliminary analysis of calcium recordings was carried out using Image J Software (public domain software developed at the US NIH). Minor drift in the XY plane of image stacks was post hoc corrected using TurboReg (Image J plugin). For each slice recorded, 50 ROIs of 15 μm² were selected for analysis in AcbC and 50 ROIs in AcbSh, and average fluorescence values within each ROI were calculated. MATLAB software (R2018a; Mathworks, Natick, MA, USA) was used to analyze Ca²⁺ parameters from the raw values. The first 5 s of the ROI signals was removed (corresponding to highly non-linear bleaching) and then the signals were low-pass filtered (Chebyshev type II). Each signal was discretized in segments of a fixed time length, and the 80th percentile of each segment was found. A line was fitted to the found points and was defined as the basal line ($F_0$). Each point of each ROI was expressed as the absolute change ($|\Delta F|/F_0|$) between the ROI point and the corresponding point in the basal line of that ROI.

An event threshold (event_th) for each ROI was calculated as

$$event\ th_p = \frac{3}{n-1} \sum_{i=2}^{n} |x_{pi} - x_{pi-1}|$$

where $x_{pi}$ is the $i$th point of the $p$th ROI signal. Any event threshold below 0.004 (0.4% change) was automatically set to that value to avoid false positives in silent signals due to stochastic noise. Event peaks were defined as the local maxima that are above the given event threshold with a prominence of at least 20% the threshold value, and that are separated from the peak of any other event for at least 3 s. These local maxima were used to measure Ca²⁺ spike amplitude ($\Delta F/F_0$), and the number of events per minute was calculated for Ca²⁺ spike frequency. MATLAB's built-in function "findpeaks" was used for the event detection step. Average responding ROIs (%), Ca²⁺ spike frequency (min⁻¹) and Ca²⁺ spike amplitude ($\Delta F/F_0$) were calculated for 3 min before and 3 min after afferent optostimulation. Each stimulated slice response was compared and normalized to its basal-activity control recording. Temporal analysis of calcium activity within this 3 min was determined by grouping Ca²⁺ spike frequency (min⁻¹) data in 20 s bins and expressing them vs time (min).

To measure CaMPARI$_{GFAP}$ AAV-transfection in the NAc (Supplementary Fig. S13E), frame 10 of basal recording was used as representative image of viral expression. ROIs were delimited manually for AcbC and AcbSh and fluorescence intensity ($\Delta F/F_0$) was calculated for each subregion, being $F_0$ a background ROI located in a region outside the NAc with no expression. Final transfection degree was determined as the average of the slice's fluorescence ($\Delta F/F_0$) considered for that hemisphere.

## CaMPARI$_{GFAP}$ photoconversion

NAc slices (300-μm thick) with astrocytes expressing CaMPARI$_{GFAP}$ and glutamatergic fibers expressing ChR2-EYFP were placed into a microscope recording chamber (Olympus BX50WI; Olympus Optical, Tokyo, Japan) superfused with gassed ACSF in the presence of picrotoxin (0.05 mM). After 5 min to allow picrotoxin perfusion, light stimulation protocols were applied: (1) optogenetic stimulation protocol followed by photoconversion protocol (40 s of violet light (λ = 405 nm; Thorlabs M405F1)) or (2) basal condition, in which only photoconversion protocol was applied. Full-field ChR2 optogenetic stimulation was delivered using CoolLED's pE-300white system through the microscope at a fixed 70% light intensity (λ = 470 nm; 70 mW/cm²) using a ×10 microscope objective. Violet light (405 nm) was delivered obliquely at a fixed intensity (1–5 mW/cm²) and did not cause tissue damage[38–40], all slices were placed at the same distance and orientation to the light beam in order to avoid variation in the scattering pattern. NAc consecutive slices at coordinates AP + 1.3 mm and AP + 0.98 mm were used for basal and optostimulation protocols, being paired for posterior analysis. These AP coordinates were alterned between basal-optostimulation in different hemispheres. Terminal NAc slices (AP + 1.6 mm and +0.7 mm) were not included since at these coordinates reliable spatial alignment between samples cannot be achieved. To minimize experimental variability, basal-optostimulated pairs underwent in parallel fixation and image collection processes. Slices were fixed with 4% paraformaldehyde (PFA), 4% sucrose in PBS for 1 h at RT. After fixation, slices mounted in Vectashield antifading mounting medium (Vector Laboratories, Burlingame, CA). Images were acquired with a Leica SP-5 inverted confocal microscope using Leica LAS AF software. Z-stack (10 plane/10 μm thickness) mosaics were collected for the three fluorescence signals: CaMPARI$_{GFAP}$ green (488 nm excitation laser), EYFP (510 nm excitation laser) and CaMPARI$_{GFAP}$ red (561 nm excitation laser). To compare fluorescence intensity between different samples, all images were acquired under identical conditions. Since photoconversion is robust in astrocytes, the decrease of CaMPARI$_{GFAP}$[27,28] did not compromise red signal detection, and no posterior immunostaining was needed.

To measure CaMPARI$_{GFAP}$ AAV-transfection in the NAc (Supplementary Fig. S13F), confocal images were used. ROIs were delimited manually for AcbC and AcbSh and the average z-stack fluorescence ($\Delta F/F_0$) was calculated for each subregion, being $F_0$ a background ROI

located in a region outside the NAc showing no viral expression. Final transfection degree ($\Delta F/F_0$) was determined as the average of the pair of slices considered for that hemisphere.

## Partition in regular quadrants (PRQ)

Image pre-processing was performed in the following manner: all CaMPARI$_{GFAP}$ red confocal images were registered to a reference mask using bUnwarpJ (Image J plugin) to align the same anatomical location across them. MATLAB software (R2018a; Mathworks, Natick, MA, USA) was used for fluorescence analysis. An average image from each stack was calculated, and images were divided into $50\,\mu m \times 50\,\mu m$ square grids. For each square grid, we calculated an individual value as the mean fluorescence signal (what we referred to as pixel). A reference autofluorescence signal for each grid was computed as the mean signal of that grid in control images ($n = 9$ slices, 3 mice) in which no virus was injected. This reference was subtracted from the images analyzed to account for the heterogeneity in the autofluorescence signal across the NAc. Moreover, each image was normalized to its background signal computed from an ROI located outside of the NAc. That is,

$$G_{i,j} = \frac{\sum\limits_{i,j} F - R_{i,j}}{F_0}$$

where $G_{i,j}$ is the value assigned to grid, $\sum_{i,j} F$ corresponds to the mean fluorescence signal inside the grid, $R_{i,j}$ is the reference autofluorescence signal of grid, and $F_0$ is the background signal outside the NAc of the image (Supplementary Fig. S6). Lastly, to compare basal-optostimulated CaMPARI$_{GFAP}$ pairs of images, each pair was normalized to the mean fluorescence in the NAc in the basal condition. Results were expressed as CaMPARI$_{Red}$ fluorescence (arb.u.) change from basal. All the custom code used for PRQ spatial analysis is available at https://github.com/JulioEI/PRQ.

## Spatial analysis

To measure fluorescent changes across space, two lines were depicted in the CaMPARI$_{GFAP}$ red PRQ images starting from dorsal regions and following nucleus anatomy (see as example Fig. 3C, D). Fluorescence quantification (arb.u.) within these lines shows signal variation from distance 0 until the complete line's length; 25 pixels ($1250\,\mu m$) for AcbC and 44 pixels ($2200\,\mu m$) for AcbSh. Astrocytic activity quantification was determined as the average line value of CaMPARI$_{Red}$ fluorescence (arb.u.) in each subregion. Representative images were truncated between the values 0.8–2 arb.u. to visually unmask the existing differences.

To analyze spatial differences between afferents and astrocytic response, an activation threshold based on fluorescence signal was used to create binary activation masks from the PRQ images (CaMPARI$_{GFAP}$ red and glutamatergic or dopaminergic afferents) by defining pixels as active (1, ≥ threshold) or inactive (0, < threshold). The activation thresholds were determined automatically by a k-mean clustering method using $k = 5$ clusters for glutamatergic and dopaminergic afferents, and $k = 6$ clusters for CaMPARI$_{GFAP}$ red images (Supplementary Fig. S7). Afferent-astrocyte paired activation masks were used to determine spatial colocalization areas (% overlap). Area relationships between overlap/astrocytes ($x$ axis) and overlap/glutamatergic afferents ($y$ axis) were tested between AcbC and AcbSh subregions (In the right of Figs. 3F, 5F, 7F, and Supplementary Fig. S12F); values close to 0 indicate no overlap while values close to 1 indicate that all of the afferent's area overlaps. Bivariate representation of the two allows dissection of the spatial distribution of these overlapped areas; values close to the diagonal (45°) indicate that the overlap area is located equally between afferents and astrocytes areas, while values close to the $x$ or $y$ axis indicates that colocalization is embedded respectively within one of the two. Multivariate analysis of variance (MANOVA) was used to determine differences between AcbC and AcbSh distributions.

To study spatial pattern differences among different pathways, a spatial correlation approach was used. For astrocytic activity analysis (Fig. 8A and Supplementary Fig. S10), only pixels above the activation threshold were considered. Using these data, average PRQ images containing spatial information of the astrocytic active regions were generated, showing each pixel CaMPARI$_{Red}$ fluorescence (arb.u.) filtered values. Pixel-by-pixel Pearson correlation was calculated by pairing fluorescence values from pixels occupying same space from different conditions and running a Pearson correlation test. For overlap domain analysis, overlap binary masks containing colocalization pixels between glutamatergic afferents and astrocytic binary masks were calculated for each individual slice. Average PRQ images were generated obtaining NAc probability maps in which each pixel value is concealed within 0 and 1 and show the overlap probability in a specific space (Fig. 8C). All the custom codes used are available at https://github.com/JulioEI/PRQ.

## Immunohistochemistry

C57BL/6 wild-type mice transfected with CaMPARI$_{GFAP}$ viral vector in the NAc were euthanized with sodium pentobarbital and transcardially perfused with PBS followed by ice-cold 4% paraformaldehyde (PFA) and 4% sucrose in PBS. Brains were removed and postfixed overnight (o/n) in the same fixative solution. Coronal brain slices of $50\,\mu m$ were obtained in a VT100S vibratome (Leica). Slices were permeabilized with 1% Triton in PBS, and non-specific binding was blocked for 1 h at room temperature (RT) with 0.3% goat serum, 0.1% Triton in PBS. After blocking, to assess CaMPARI$_{GFAP}$ specificity, sections were incubated with the corresponding primary antibodies at 4 °C o/n, and with secondary antibodies for 1 h at RT. The primary antibodies used were rabbit anti-S100 (1:200; ab868, Abcam) and mouse anti-NeuN (1:500; MAB377, Merck). Secondary antibodies were goat anti-mouse Alexa 647 (1:1000; A21236, Invitrogen) and goat anti-rabbit Alexa 405 (1:1000; A31556, Invitrogen). To quantify astrocytic density in the NAc and test 405 nm light cytotoxicity, astrocytes and microglia were labeled with primary antibody rabbit anti-S100β (1:1000; 287003, Synaptic Systems), mouse anti-GFAP (1:400; G3893, Sigma) and guinea pig anti-Iba1 (1:500; 234004, Synaptic Systems). Secondary antibody goat anti-rabbit Alexa 405 (1:1000, A31556, Invitrogen), goat anti-mouse Alexa 488 (1:1000, A11001, Invitrogen) and goat anti-guinea pig Alexa 647 (1:1000, A21450, Invitrogen). Finally, sections were washed with 0.1% Triton in PBS and mounted for imaging in Vectashield anti-fading mounting medium (Vector Laboratories, Burlingame, CA). Fluorescence images were acquired with a Leica SP-5 inverted confocal microscope using Leica LAS AF software. Image analysis was carried out with Image J software (public domain software developed at the US NIH) and Cell Profiler software was used for automatized cell quantification.

## Drugs and chemicals

Picrotoxin (#1128), 6-cyano-7-nitroquinoxaline-2,3-dione (CNQX; #0190), D-(-)-2-amino-5-phosphonopentanoic acid (D-AP5; #0106), 2-methyl-6-(phenylethynyl)pyridine hydrochloride (MPEP; #1212), ($R$)-( + )−7-Chloro-8-hydroxy-3-methyl-1-phenyl-2,3,4,5-tetrahydro-1$H$−3-benzazepine hydrochloride (SCH 23390; #0925), 4-[4-(4-chlorophenyl)−4-hydroxy-1-piperidinyl]−1-(4-fluorophenyl)−1-butanone hydrochloride (Haloperidol; #0931), (4 $R$,4a$R$,5 $R$,7 $S$,9 $S$,10 $S$,10a$R$,11 $S$,12 $S$)-octahydro-12-(hydroxymethyl)−2-imino-5,9:7,10a-dimethano-10aH-[1,3]dioxocino[6,5-d]pyrimidine-4,7,10,11,12-pentol citrate (TTX; #1069), (3 $S$,3a$R$,4 $S$,6 $S$,6A$R$,7 $S$,8 $S$,9b$S$)−6-(acetyloxy)−2,3,3a,4,5,6,6a,7,8,9b-decahydro-3,3a-dihydroxy-3,6,9-trimethyl-8-[[(2Z)−2-methyl-1-oxo-2-butenyl]oxy]−2-oxo-4-(1-oxobutoxy)azuleno[4,5-b]furan-7-yl octanoate (Thapsigargin; #1138) and 1,2-bis(2-aminophenoxy)ethane-N,N,N′,N′-tetraacetate (BAPTA; #2786) were purchased from Tocris. (S)−3,5-DHPG, group I mGlu agonist (#ab120007) was purchased from Abcam. Adenosine 5′-triphosphate disodium salt hydrate (#A7699) and Nε-

(+)-Biotinyl-L-lysine, Bct (Biocytin; #B4261) were purchased from Sigma-Aldrich.

## Statistical analysis

All data are expressed as mean ± SEM. Statistical analysis of the differences between groups were determined by two-tailed unpaired or paired $t$ test, one-way ANOVA or two-way ANOVA followed by Holm–Sidak test for multiple comparisons, unless otherwise indicated. Differences were considered to be significantly different when $P < 0.05$. Significant changes from basal, in the case of normalized data (basal = 1), were determined by one-sample $t$ test. Correlation analysis was performed by Pearson's test. Statistical differences among XY distributions were determined by multivariate analysis of variance (MANOVA). Statistical differences were calculated using GraphPad Prism 7 and MATLAB (R2018a; Mathworks, Natick, MA, USA). All statistical details are provided at Supplementary Table 1.

## Reporting summary

Further information on research design is available in the Nature Research Reporting Summary linked to this article.

## Data availability

All data generated in this study are provided in the Supplementary Information and Source Data file. The raw data images that support the findings of the current study are available from the corresponding author upon reasonable request.  Source data are provided with this paper.

## Code availability

All custom codes used for Partition in Regular Quadrants (PRQ) spatial analysis (Figs. 3, 5, 7, 8, 9, and Supplementary Figs. S6–S9 and S12) are available at Github: https://github.com/JulioEI/PRQ

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

## Acknowledgements

We thank Alfonso Araque, Eduardo Martín, Juliana M Rosa, and Gertrudis Perea for expert advice and critical reading of the manuscript. This work was supported by grants from the Spanish Ministry of Science and Innovation (Ramón y Cajal RYC-2016-20414, RTI2018-094887-B-I00, and PID2021-122586NB-I00) to M.N. and Fondo Europeo de Desarrollo Regional (FEDER) and PID2020-115091RB-10 to R.T. The professional editing service NB Revisions was used for technical preparation of the text prior to submission.

## Author contributions

I.S. carried out most of the experimental work and participated in manuscript writing. J.E. developed PRQ analysis method. C.M.-M. carried out some immunohistochemical analysis. L.D., P.P., M.P., and R.T.

helped in the analysis of data. M.N. designed the research and wrote the paper.

## Competing interests

The authors declare no competing interests.
