## [Peer Review File · Nature Communications]

Ca²⁺-modulated Photoactivatable Imaging Reveals Neuron-Astrocyte Glutamatergic Circuitries within the Nucleus AccumbensReviewers' comments:

Reviewer #1 (Remarks to the Author):

The very interesting work by Serra et al. uses a novel methodology (adapting an existing tool) to evaluate astrocytic activation (GFAP-CAMPARI), and performs functional imaging of nucleus accumbens astrocytes in response to optostimulation of different glutamatergic afferents (mPFC, BLA and vHIP). They show input-specific astrocytic driven responses induced by optogenetic activation of different inputs. The work is original, and is of wide interest for the field. A large focus has been given to the functional specificity of distinct NAc subcircuits but mostly in terms of neuronal responses. This study now suggests that there are also specific patterns of astrocytic activity in response to stimulation of different glutamatergic inputs. The experiments were conducted in an appropriate manner, and the methodology is adequate. However, since the manuscript uses a novel methodology to evaluate astrocytic activation maps, I have some technical concerns that need to be explained by the authors, as I will detail below.

Major concerns:

1. To show specificity of the approach, as a control experiment, authors should perform the same type of experiment depicted in Fig1E-F, but with another neurotransmitter that does not induce a response in astrocytes.
2. Considering that a large proportion of the manuscript is based on expression signals of glutamatergic terminals and CAMPARI signal in astrocytes, it is very important to control for degree of transfection both in input regions (fluorescent reporters for mPFC, amy and vHip) but also in the NAc (CAMPARI). From the methodology it was not clear to me if this control was made, and this is crucial considering the large degree of variability of transfection between animals even using the same volume and titer of virus, injection coordinates etc.
3. A note regarding the dynamic range of astrocytic activation – did authors test this? Because in some of the activity images, it appears that the signal is quite saturated?
4. Again related to the methodology: Authors showed the astrocytic activity in response to a particular optical stimulation protocol of a glutamatergic input. What was the basis to choose these parameters for optical stim? Is the astrocytic response the same to different stimulation conditions? Can this methodology detect different activation masks depending on different stimulation conditions of the same input? Is the methodology sensitive to detect changes in astrocytic activity due to different input activation?
5. What is the explanation/hypothesis behind the effects of co-stimulation of 2 or 3 inputs in comparison to 1 input stimulation? Authors explain that the astrocytes integrate information from different inputs, and I do agree with this but how do you explain that the sum of glutamatergic inputs can even decrease astrocytic activity? What is behind this? – this should be discussed.
6. For me it was confusing to understand for how long the astrocytic calcium recordings were done. For example in Fig1C3 was it 3 min? Considering this time frame, then the activity reflects i) a direct effect of glutamatergic inputs activation of these astrocytes, ii) indirect effect. Thus, the study would benefit from a temporal analysis of this signal, i.e., is the activity change of the first seconds similar to the last seconds of the 3 minutes? What are the immediate vs delayed responses?
7. Some graphs are really difficult to interpret because axis legends are incomplete or legends. I suggest revising all the figures to ensure that proper axis and information is provided in the legend to interpret the data of the figure.
8. The increase in mit copies in shell appears to be explained by a smaller fraction of the astrocytes in comparison to the large portion that are similar between core and shell. This should be discussed.

Other comments:

9. It is not clear if the whole NAc region was analyzed – which were the most anterior and most posterior sections used for analysis? As described in line 564?
10. Last section of results, the figure number is mistaken through the text, it should be Figure 8
11. Since there are 2 versions of CAMPARI, authors should refer to which version they are using in this manuscript.
12. Scale of Sup fig 1 is not clear? What exactly is shown?
13. It was not clear how data from Fig. 7 was calculated? What are the axes?

14. Optogenetic parameters should be given in detail, some important parameters are missing
15. The method for evaluation of anatomical density of projections should be better explained in the material and methods section – from line 492 on on a separate section with more details. How many slices, how many animals? What was used as a control ROI region?
16. In a similar note, the PRQ method should also be described in a separate section
17. Regarding the same topic as above, authors should also clearly state when Chr2 or Chrimson was used, it is not clear throughout the manuscript.
18. Line 126 remove “medium spiny neurons” because you are evaluating the whole region, not specifically these neurons (for anatomical experiments)
19. The use of GABA A receptor blockers in the recording medium should be properly discussed, especially considering the neuronal constitution of the NAc, which is mostly GABAergic neurons and interneurons.
20. Sup fig 3 -What data supports the use of this method to observe CA2+ spike amplitude? Does it have resolution to do so?
21. Line 155-156 – remove opposite; use different
22. MPEP experiment: the responses depicted in the graphs refer to core or shell? I assume is shell?
23. Maybe it is because I am not familiarized with this methodology, but Fig. 2D3 is not clear to me. What do you want to depict? Distance from?
24. I suggest that the mitochondrial data is provided in a distinct section
25. Legends fig 3e3 missing, same in 4, same in 5C1
26. Scale in fig4c2 is missing
27. Line 301, 325, 422-423, not clear
28. Describe data of fig6a and b in results text
29. Line 441 – co-stimulated responses should be changed to stimulation of 2 or 3 inputs simultaneously ...
30. The methodology could be better explained; for example it is not clear when authors present CaMPARI or photoconverted CaMPARI data.
31. the graphs axis are not always clear to interpret what is represented. Maybe add more information to the results section or legends?
32. P values of Pearson correlations are missing in legends
33. Line 352-354 – I think is the other way around?

Personal notes:

- I found really difficult to follow figures having E1, E2, D1, D2, D3, D4 etc, I suggest using exclusive letters for each section of figures
- The mit data is interesting but, in my modest opinion, as it is, does not add much to the main findings.

Ana João Rodrigues

Reviewer #2 (Remarks to the Author):

Investigation of pathway-specific activation of glutamatergic fibers to the nucleus accumbens (NAc) placed the distinct neuronal input and/or output features in the centre. Therefore, conjuring up the role for distinguishable astrocyte assemblies integrating pathway-specific glutamatergic transmission is certainly an important and sensible approach [Kardos et al., Molecular plasticity of the nucleus accumbens revisited - astrocytic waves shall rise; Mol. Neurobiol. 2019, 56, 7950-7965; doi: 10.1007/s12035-019-1641-z.]. In order to study functional neuron-astrocyte circuitries in the NAc, Authors devised a fluorescent technique based on calcium-modulated photoactivatable ratiometric integrator (CaMPARI) that undergoes irreversible green-to-red fluorescence conversion upon coincident elevated intracellular Ca(2+) and ultraviolet light illumination [Fosque et al., Neural circuits. Labeling of active neural circuits in vivo with designed calcium integrators; Science 2015, 347, 755-760; doi: 10.1126/science.1260922]. Authors claim that the selective optostimulation of main glutamatergic inputs (i.e. prefrontal cortex, basolateral amygdala and ventral hippocampus) induces astrocytic Ca(2+) activities mediated by metabotropic glutamate receptor mGluR5 that do not coincide with glutamatergic innervations, suggesting unexpected

neuron-astrocyte circuitries. Interestingly, the differences in basal Ca(2+) dynamics between the NAc shell and core astrocytes were associated with differences in mitochondrial DNA copy number, exhibiting molecular heterogeneity in the regulation of their mitochondrial genomes.

Unfortunately, the description of the novel approach applied to monitor astrocyte activity (CaMPARIGFAP) is not adequately addressed at present. My specific questions (Qs) are as follows:

1. Why should the elevation of intracellular Ca(2+) by ATP decrease the CaMPARIGFAP fluorescence in astrocytes?
2. Why does the ATP-induced decrease in CaMPARIGFAP fluorescence demonstrate "the ability of the molecule to monitor Ca(2+) dynamics"?
3. Authors claim the application of UV light during a fixed temporal window, however, they do not explicate why. In reality, UV light may seriously affect cellular viability conjuring up the question how does the fixed temporal window ensure that this would not be happening under the explicit application protocol?
4. How do we know that the green-to-red photoconversion occurred in those astrocytes that were active at the moment of illumination if ATP stimulation decreases the CaMPARIGFAP fluorescence (c.f. Q1 and Q2)?
5. My understanding is that the Red/Green fluorescence ratio (FRed/FGreen) of CaMPARIGFAP changes according to the distance from the site of ATP application. The application distance, however, may vary from one experiment to another. I assume that this kind of variability does impact the quantification of Ca(2+) transients as well. Furthermore, in order to get unconditional data, the application of the CaMPARIGFAP tool necessitates devising relative data such as for example (FACTUAL-FBASAL)/FBASAL.
6. The applied confocal microscopy may suffer from out-of-focus light contaminating the region of interest, and thus accurate green fluorescence measurements are not possible in the presence of non-specific background fluorescence, interfering the generation of an accurate FRed/FGreen ratio.
7. As with any activity reporter, a critical aspect of interpreting experiments using CaMPARI is to carefully calibrate readout versus the underlying phenomenon under study; importantly, such calibrations should take place in the preparation of interest (when possible), as factors such as expression level, long-term stability, and light delivery and collection can vary widely. [c.f. Zolnik et al., All-optical functional synaptic connectivity mapping in acute brain slices using the calcium integrator CaMPARI. *J. Physiol.* 2017, 595, 1465-1477; doi: 10.1113/JP273116]

Reviewer #3 (Remarks to the Author):

In this manuscript, Serra and colleagues dissected the functional connectivity between distinct glutamatergic neuronal circuits and astrocyte networks in the nucleus accumbens (NAc). To study the neuron-astrocyte interaction in NAc, the authors 'smartly' combined optogenetic stimulation of neurons with the optical monitoring of astrocyte-network activity using genetically encoded Ca²⁺ activity integrator (CaMPARI). The authors used recombinant AAVs to express channelrhodopsin2 (Chr2) in the neurons in various brain areas differentially projecting to the NAc core (AcbC) and shell (AcbSh) region of NAc. These brain regions include the medial prefrontal cortex (mPFC), amygdala, ventral hippocampus (vHip), and ventral tegmental area (VTA). By simultaneously activating neuronal Chr2 (with blue light) and astrocytic CaMPARI (UV-light), the authors tried to capture the astrocyte network activated in response to neuronal activation. As proof of principle, the authors combined optogenetics and electrophysiology to show a strong positive correlation between glutamatergic fiber innervation and medium spiny neurons (MSNs) responses in the core and shell of NAc. On the contrary, astrocyte network activation (i.e., intracellular Ca²⁺ increase) doesn't correlate with the glutamatergic fiber innervation and neuronal activity patterns. Although the inverse relationship between neurons activation and astrocyte network activity is fascinating, the authors don't provide any potential mechanism driving the differential response of astrocytes on activation of individual glutamatergic pathways. In the current state, the manuscript requires a significant body of work for conveniently establishing the significance of these observations.

Major concerns:

1. The authors perform all their CaMPARI experiments in the presence of picrotoxin, quite likely to

block the action of inhibitory neurons. Although this is critical information, the authors don't explicitly mention this in the result section and discuss the rationale behind using picrotoxin. Why do they need to block inhibition in their brain slices? This can already induce neuronal activation in the absence of any optogenetic stimulation.

2. The authors show that astrocytes Ca^{2+} increase is more robust in regions with less glutamatergic fiber innervation (see Fig 2. B3/C2 and Fig 4. B3/C2) and is dependent on mGluR5 activation. If NAc astrocytes activation is dependent on glutamate release, what is the source of this glutamate? If the source is synaptic glutamate release, we should expect an overlap between glutamatergic neuronal innervation and astrocyte activation.

3. From Fig. 2C1 (and 2E2), it seems AAV-based expression of CaMPARI in non-homogenous across NAc, i.e., more astrocytes express CaMPARI in shell than the core. Hence, the difference in the astrocyte activity and the innervation profile of afferents from various brain regions probed in this study can simply emerge from the number of astrocytes expressing CaMPARI in shell vs. core.

4. In Figure 4, there is a discrepancy between C2 and E2. Based on the images shown, there is a reasonably strong activation of AcbSh astrocytes (E2), but in C2, this seems to be relatively mild. Also, when vHip afferents are optogenetically stimulated, astrocytes in the entire area covered by these afferents show a Ca^{2+} increase (Fig. 4F1). At least, in this region, the reverse correlation between neuronal and astrocytic activity is not as evident as seen for mPFC and Amygdala (compare figures 2F1, 3F1, and 4F1).

5. In general, AcbSh astrocytes always respond to optogenetic stimulation of glutamatergic afferents (see figures 2-4 E2), and when afferents from vHip are stimulated, then astrocytes in AcbC also strongly responded. Hence, from this point of view, it will be difficult to conclude that there is any specific co-relation between glutamatergic afferents innervation and astrocyte activity.

6. How does the variability in the mitochondrial DNA (mtDNA) copy number (Fig.6B1, B2) relate to the differential astrocytic Ca^{2+} response seen between AcbSh and AcbC (Fig. 6B2)? What is the source of a large scatter in the mtDNA/cell in AcbC (Fig. 6B2)? Most of the cells have similar mtDNA copies in AcbC and AcbSh (Fig. 6B2); few AcbC cells show a considerable variation in the mtDNA copy number. In short, the key question is, how does differential mitochondrial DNA copy number in astrocytes allow for differential input integration of glutamatergic pathways by astrocytes?

7. Figure 7 is entirely confusing, and it is unclear what message the authors want to convey. The plots presented in this figure are non-intuitive and show contrary information to those shown before in Fig. 2, 3, and 4. Also, there is some level of redundancy between figures 5 and 7.

8. What is the mechanism by which the activation of the amygdala can suppress the activation by vHip (Fig. 8B1, D1), given that mPFC doesn't suppress the activation of vHip (Fig. 8C1)? Why would co-stimulation of all pathways (mPFC, amygdala, and vHip) don't induce Ca^{2+} transients (Fig. 8D1-D3), given that individual pathways activation-induced Ca^{2+} transients in AcbSh and AcbC? The authors don't provide any mechanism behind this crucial observation.

Minor concerns:

- Line 887 – Fig. 2D4 is related to the neuronal afferents and not the astrocyte CaMPARI signal.
- What does $\Delta F/F_0$ represent in Fig. A2? There is no concept of time in this image.
- Typo: the last section should be Fig. 8, but Fig. 7 is mentioned throughout the text.

We would like to thank the reviewers for the interesting and helpful comments that have helped to improve the results and strengthen the conclusions. The suggestions from the reviewers to improve the manuscript are all very helpful. Therefore, we have followed all suggestions, making for a better report of our findings. Below is a point-by-point description of how we addressed/will address each of the reviewers' suggestions.

(Reviewer comments are in *blue italics*; sentences included in the manuscript are in *italics*).

Reviewer #1:

The very interesting work by Serra et al. uses a novel methodology (adapting an existing tool) to evaluate astrocytic activation (GFAP-CAMPARI), and performs functional imaging of nucleus accumbens astrocytes in response to optostimulation of different glutamatergic afferents (mPFC, BLA and vHIP). They show input-specific astrocytic driven responses induced by optogenetic activation of different inputs. The work is original, and is of wide interest for the field.

A large focus has been given to the functional specificity of distinct NAc subcircuits but mostly in terms of neuronal responses. This study now suggests that there is also specific patterns of astrocytic activity in response to stimulation of different glutamatergic inputs. The experiments were conducted in an appropriate manner, and the methodology is adequate.

However, since the manuscript uses a novel methodology to evaluate astrocytic activation maps, I have some technical concerns that need to be explained by the authors, as I will detail below.

We thank the reviewer for the supportive comments regarding the work's originality, the recognition of the amount of data collected, and the helpful suggestions that allowed the clarification of important issues.

Major concerns:

1. To show specificity of the approach, as a control experiment, authors should perform the same type of experiment depicted in Fig1E-F, but with another neurotransmitter that does not induce a response in astrocytes.

As described in more detail below, we have carried out a new set of experiments to validate the molecule as a calcium sensor as suggested by the reviewer. We have used 1) Thapsigargin (1 μ M), which depletes intracellular Ca^{2+} stores by inhibiting Ca^{2+} ATPase (Navarrete and Araque, 2008; Perea and Araque, 2005). In Figure 1C₄, after perfusing slices with thapsigargin for 30-45 min, which depletes internal stores by inhibiting Ca^{2+} ATPase (Araque et al., 1998; Navarrete and Araque, 2008; Perea and Araque, 2005), these calcium oscillations in response to ATP were abolished (1.67 ± 1.67 % in thapsigargin vs. 98.9 ± 1.11 % in control; $n = 6$ thapsigargin slices and $n = 9$ control slices; $P < 0.001$) indicating that they were mediated by calcium release from intracellular calcium stores. 2) Intracellular infusion of the calcium chelator BAPTA (20 mM) into astrocytes using a patch pipette, to specifically chelate Ca^{2+} in the astrocyte network (Baudon et al., 2022, Navarrete et al., 2019). It is well known that BAPTA propagates through gap junctions in the astrocytic syncytium, interfering with astrocytic Ca^{2+} signaling throughout the slice (Navarrete and Araque, 2010, Jourdain et al, 2007, Baudon et al., 2022, Navarrete et al, 2019). See new Supplementary Figure S1 for a representative example of intracellular loading of astrocytes with BAPTA and biocytin, followed by streptavidin-Alexa 647 staining, and for widespread quenching of astrocytic Ca^{2+} signals after intracellular loading with BAPTA. In Figure 1C₄, Ca^{2+} oscillations in response to ATP were also abolished after BAPTA infusion into the astrocytic syncytium ($n = 2$ slices; $P < 0.001$).

In parallel, calcium signaling was assessed by measuring $\text{CaMPARI}_{\text{GFAP}}$ photoconversion in

response to ATP (new Fig. 1D). We show that astrocytic calcium activity evoked by local ATP puff is revealed by an increase in red fluorescence in the control condition, whereas in the presence of Thapsigargin or BAPTA this signaling is abolished (Figure 1D₂). This set of experiments confirm that CaMPARI_{GFAP} is a functional calcium indicator for monitoring astrocyte Ca²⁺ dynamics both in real-time (new Fig. 1C and new Figure S1) and after photoconversion of the molecule (new Fig. 1D).

These data have been included in the text, in 4 new panels on Figure 1 and new Supplemental Figure 1 have been added (P. 5, l. 5).

2. Considering that a large proportion of the manuscript is based on expression signals of glutamatergic terminals and CaMPARI signal in astrocytes, it is very important to control for degree of transfection both in input regions (fluorescent reporters for mPFC, amy and vHip) but also in the NAc (CaMPARI). From the methodology it was not clear to me if this control was made, and this is crucial considering the large degree of variability of transfection between animals even using the same volume and titer of virus, injection coordinates etc.

As indicated by the reviewer, we have included as a new Figure S12 the plots of transfection degrees (i.e. the glutamatergic opsins and CaMPARI_{GFAP} injections). Analysis of these data shows no significant differences in transfection variability.

Furthermore, the analysis method has been included in detail in a new section “**Afferent density and opsin transfection**” (P. 22, l. 17) “*Injection site and NAc slices were kept and fixed with 4% paraformaldehyde (PFA), 4% sucrose in PBS for 1 h at RT. After fixation, the slices were incubated with DAPI (1.5 µg/ml, Sigma-Aldrich) for 10-15 min and all were mounted in Vectashield antifading mounting medium (Vector Laboratories, Burlingame, CA). Fluorescence images were acquired with a 10x objective in a Leica AF 6500-7000 microscope using Leica LAS AF software. Image analysis was carried out with Image J software (public domain software developed at the US National Institutes of Health [NIH]). Regions of interest (ROIs) were delimited manually in each slice, and average fluorescence inside the regions was expressed as the change ($\Delta F/F_0$) between the averaged ROI value (F) and a background ROI (F_0) set in a reference point of the tissue without reporter’s fluorescent signal. To measure AAV-transfection at the injection site (Fig. S12A), same number of AP slices (300 µm thick) were analyzed in each nuclei: mPFC (AP: 2.1 mm to 1.6 mm), Amyg (AP: -1.2 mm to 2.4 mm), vHip (AP: -2.8 mm to 4 mm) and VTA (-3 mm to -3.9 mm). Final transfection degree was calculated as the average fluorescence ($\Delta F/F_0$) of the set of slices. To measure afferent’s density in the NAc, ROIs were delimited manually for AcbC and AcbSh and fluorescence intensity ($\Delta F/F_0$) was calculated for each subregion, being F_0 the background ROI located in a region outside the NAc with no afferent’s innervation. Final afferents fluorescence (a.u.) was expressed as the individual AcbC and AcbSh ($\Delta F/F_0$) values relative to the transfection degree ($\Delta F/F_0$) measured at their respective injection site.”*

3. A note regarding the dynamic range of astrocytic activation – did authors tested this? Because in some of the activity images, it appears that the signal is quite saturated?

We thank the reviewer for this remark, we have not determined nor observed an upper limit in our analysis regarding the dynamic range of astrocytic activation, which goes from basal activity (1 a.u.) to different levels of increase in the signal that can be observed in the bar chart quantifications (Fig. 2E4, 3E4, 4E4, 5A2, 7A3, 7B3, 7C3, 7D3, S8E, S9C3 and S11D4). The reason for which activity images can become saturated is that the signal is restrained within a truncated range between (0.8 - 2 a.u.), which was consciously selected to highlight differences maintaining the same color scale among experimental conditions as shown in the colormaps index. To clarify the issue, we have included the following paragraph in material and methods “Representative images were truncated between the values 0.8 - 2 a.u. to

visually unmask the existing differences.” (P. 28, l. 16)

4. Again related to the methodology: Authors showed the astrocytic activity in response to a particular optical stimulation protocol of a glutamatergic input. What was the basis to choose these parameters for optical stim? Is the astrocytic response the same to different stimulation conditions? Can this methodology detect different activation masks depending on different stimulation conditions of the same input? Is the methodology sensitive to detect changes in astrocytic activity due to different input activation?

We thank the reviewer to raise these relevant questions. Our optostimulation protocol (10 pulses 50 ms at 4Hz - 4 times, 5 s interval) was designed based on experimental evidence from Mattis et al., 2011 and Britt et al., 2012 studies, with the aim of stimulating the opsins expressed at the glutamatergic afferents in a reliable way. This protocol at 4 Hz remained constant for all experiments to allow for astrocytic activity comparison between pathways. Furthermore, considering our resulting dose-response curve (see Fig. S4B), we chose the final stimulation light intensity: 70 % (P. 24, l. 2). As can be seen, this light intensity is within the plateau curve of EPSCs.

Additionally, to address the reviewer’s questions and further explore the detection capabilities of our methodology, we have performed a new experiment using a different optostimulation protocol at 100 Hz, which is being reported to optogenetically induced LTP at the vHip-AcbSh pathway (LeGates et al. 2018). As in previous experiments (Fig. 2E₄, 3E₄, 4E₄, 5A), ChR2 expressed in ventral hippocampus afferents was activated in the Nucleus Accumbens and the subsequent astrocytic response was assessed using CaMPARI_{GFAP} Red signal. Our results show no differences between stimulation protocols regarding the activation area % that gathers the astrocytic increased activity (Attached Figure C), however we did find differences in the intensity of the calcium responses (Attached Figure D), suggesting that same neuron-astrocyte network is being activated at different intensity. Although the biological significance of these observations is out of the scope of our study, present results further demonstrate the sensitivity of the methodology to detect changes in astrocytic activity due to different pathway (Fig. 5A₂), or input activation (Attached Figure).

Figure. NAC astrocytic calcium activity in response to ventral hippocampus afferents at 4 Hz and 100 Hz inputs. (A) Scheme and representative coronal slices showing ChR2-EYFP expression at vHip (left; scale bar = 1 mm) and its associated axons in the NAc (right; scale bar = 500 μ m). (B) Detailed scheme of the stimulation protocols used for vHip afferent activation at 4 Hz (top) and 100 Hz (bottom). (C) Left, masks of vHip glutamatergic afferents area (green), astrocytic activation area (yellow) defined by the CaMPARI_{GAP} Red signal and the overlap area between the two (turquoise). Masks corresponding to the slices stimulated at 100 Hz. Right, quantification of the spatial overlap between vHip afferents and active astrocytes both at 4 Hz (blue bars; n = 8 slices, N = 6 mice) and at 100 Hz (turquoise bars; n = 7 slices, N = 6 mice). Two-way ANOVA, Holm-Sidak test for multiple comparisons, n.s.: p > 0.05. Note that there are no differences between masks area between stimulation conditions. (D) Left, average partition in regular quadrants (PRQ) images showing astrocytic activity above the activation threshold in response to vHip stimulation at 4 Hz and 100 Hz. Lines starting from pixel 0 in each subregion were used for quantification (pixel = 50 μ m²). Right, quantification of CaMPARI_{GAP} Red signal above threshold in response to optostimulation at 4 Hz (blue bars; n = 16 slices 8 pairs, N = 6 mice) and at 100 Hz (turquoise bars; n = 14 slices 7 pairs, N = 6 mice) (between groups; p = 0.02). Two-way ANOVA, *: p < 0.05.

5. What is the explanation/hypothesis behind the effects of co-stimulation of 2 or 3 inputs in comparison to 1 input stimulation? Authors explain that the astrocytes integrate information from different inputs, and I do agree with this but how do you explain that the sum of glutamatergic inputs can even decrease astrocytic activity? what is behind this? – this should be discussed.

We thank the reviewer for the comment. Following the reviewer's suggestion, we have discussed the issue by adding the following paragraphs (P. 19, l. 23 and P. 19, l. 5).

Our results indicate that the sum of glutamatergic inputs decreases astrocytic activity, showing the synaptic information processing by astrocytes in the nucleus accumbens. In the discussion section we introduce a new evaluation and hypothesis suggestion of which could be the underlying reasons for this. “These results agree with the reported calcium activity regulation by different synaptic inputs in the hippocampus (Perea and Araque, 2005). While further studies, out of the scope of the present work, are required to elucidate the underlying molecular mechanisms of this phenomenon, it can be hypothesized that it might be due to the interaction of the intracellular signaling pathways stimulated by both synaptic inputs (see Durkee et al., 2019; Hirrlinger and Nimmerjahn, 2022). This lack of linearity shows that the integration property orchestrated by astrocytes in the NAc (e.g., the ability shown to increase

the signal-to-noise ratio (see Mancini et al., 2021)), could mechanistically explain the divergent physiological and behavioral responses produced by the activation of different glutamatergic inputs to the NAc”

“The existence of these astrocytic networks was revealed by blocking lateral inhibition and forward inhibition using picrotoxin and preserved when neuronal activity was blocked with the Na⁺ channel blocker TTX, highlighting that defined astrocytic ensembles respond to a specific neuronal input. However, of deep interest would be to further study the implications of activating a profile of astrocytes in less-innervated areas in response to glutamatergic stimuli to enlarge the missing links of the physiological picture.”

6. For me it was confusing to understand for how long the astrocytic calcium recordings were done. For example in fig1C3 was it 3 min? Considering this time frame, then the activity reflects i) a direct effect of glutamatergic inputs activation of these astrocytes, ii) indirect effect. Thus, the study would benefit from a temporal analysis of this signal, i.e., is the activity change of the first seconds similar to the last seconds of the 3 minutes? What are the immediate vs delayed responses?

We have revised materials and methods clarifying data analysis (P. 26, l. 2): *“Average responding ROIs (%), Ca²⁺ spike frequency (min⁻¹) and Ca²⁺ spike amplitude ($\Delta F/F_0$) were calculated for 3 min before and 3 min after afferent optostimulation. Each stimulated slice response was compared and normalized to its basal activity control recording. Temporal analysis of calcium activity within these 3 min was determined by grouping Ca²⁺ spike frequency (min⁻¹) data in 20 s bins and expressing them vs time (min).”*

Furthermore, in agreement with the reviewer to better dissect astrocytic response across time, we have performed a temporal analysis of these signals, showing the Ca²⁺ spike frequency values within 20 s bins (new Fig. S5A₂, S5B₂, and S5C₂). These results are consistent with the information given by the average signals shown in figures 2C₃, 3C₃, and 4C₃, and are also in agreement with the photoconversion activity profiles reported in figures 2E₄, 3E₄, and 4E₄ in which the first 40 s after optostimulation are captured. Our hypothesis is that initial response reflects a direct activation of the astrocytic network associated with each glutamatergic input (new Fig. S8). Subsequently, we understand the activity maintained over time as a result of internal astrocytic network processing given that in the presence of neuronal activity blocker (TTX) (see new Fig. S8), at least the astrocyte response to mPFC optostimulation is similar.

7. Some graphs are really difficult to interpret because axis legends are incomplete or legends. I suggest revising all the figures to ensure that proper axis and information is provided in the legend to interpret the data of the figure.

We thank the reviewer for the comment, and we appreciate the effort in helping us to clarify the figures. We have thoroughly modified figure legends and have made some changes in the axis as follows:

- Y-axis concerning the anatomical afferents density (Fig. 2A₂, 3A₂ and 4A₂) have been corrected and an independent section providing more details about the analysis has been included in Materials and Methods (P. 22; l. 17). See also comment #15.
- Regarding CaMPARI_{GFAP} photoconversion experiments (Fig. 2E₃₋₄, 3E₃₋₄, 4E₃₋₄, 6A₅₋₆, 7A₂₋₃, 7B₂₋₃, 7C₂₋₃, 7D₂₋₃, S7A₁₋₃, S8E, S9C₂₋₃, S11D₃₋₄), y-axis has been replaced by “CaMPARI_{Red} Fluo. (a.u.)” to better indicate that astrocytic activation is measured with CaMPARI_{GFAP} Red signal.

- Same way for Fig. 2D₃₋₄, 3D₃₋₄, 4D₃₋₄ and S7B₁₋₃, y-axis has been replaced by “Glut. afferents Fluo. (a.u.)” and for Fig. S7C and S11C₃₋₄, y-axis has been replaced by “VTA afferents Fluo. (a.u.)”.
- Figure 5B (former Fig. 7A) axis and legend have been modified and further information regarding the analysis is provided in the Materials and Methods (P. 29; l. 9). See also comment #13.
- Figure S3 axis and legend have been modified. See also comment #12.
- Figure S10 (former Fig. 5B) axis and legend have been modified and further information is provided in the Materials and Methods (P. 28; l. 11).

8. The increase in mit copies in shell appears to be explained by a smaller fraction of the astrocytes in comparison to the large portion that are similar between core and shell. This should be discussed.

Our results showed enrichment of mitochondrial DNA copies in a fraction of the AcbC astrocytes collected for analysis (Fig. 6B), which is consistent with the distinct Ca²⁺ signaling observed between NAc subregions (Fig.6A). These results are in line with recent single-cell studies which explore the transcriptome in situ (Batiuk et al. 2020, Bayraktar et al. 2020, Ohlig et al. 2022), which find differential molecular and Ca²⁺ signaling between astrocytic subtypes (Batiuk et al. 2020) and also show the region-restricted mapping of genes related to mitochondrial functions, prompting the idea that metabolic specialization may be more region-specific depending on local neuronal networks (Ohlig et al. 2022). Unlike these massive sequencing techniques, our patch-dPCR provides accurate quantification of absolute mitochondrial DNA copies at a single-cell resolution which we believe is an important feature that compensates for the restricted population of cells collected due to manual sampling. Although future work is needed to further dissect molecular heterogeneity in the NAc astrocytes, our results of differential Ca²⁺ signaling in combination with mitochondrial DNA copies provide an unprecedented characterization of the nucleus in line with recent evidence reported in the field. We have included the following sentence: “*These results, in concordance with transcriptome in situ studies (Batiuk et al. 2020, Bayraktar et al. 2020, Ohlig et al. 2022 indicate an increased Ca²⁺ activity coupled with a mtDNA copy number enrichment in astrocytes. This is indeed an exciting issue, giving rise to the idea that metabolic specialization is region-specific and depend on local neuron-astrocyte circuits. However, the elucidation of the mechanisms underlying this phenomenon will likely engage extensive new research in the future*”. (P. 16, l. 7)

Other comments:

9. It is not clear if the whole NAc region was analyzed – which were the most anterior and most posterior sections used for analysis? As described in line 564?

To clarify the issue we have included the following sentence in materials and methods (P. 26, l. 24): “*NAc consecutive slices at coordinates AP +1.3 mm and AP + 0.98 mm were used for basal and optostimulation protocols, being paired for posterior analysis. These AP coordinates were alterned between basal-optostimulation in different hemispheres. Terminal NAc slices (AP + 1.6 mm and + 0.7 mm) were not included since at these coordinates reliable spatial alignment between samples cannot be achieved.*”

10. Last section of results, the figure number is mistaken through the text, it should be Figure 8

We apologize for the mistakes. We have carefully revised the present version of the manuscript.

11. Since there are 2 versions of CaMPARI, authors should refer to which version they are using in this manuscript.

We implemented for astrocytes the CaMPARI1 version engineered from Fosque et al. This information has been included in the results section (P. 4; l. 18), Figure 1A and the plasmid used is available at materials and methods *CaMPARI_{GFAP} viral vector* section (P, 21; l, 8).

12. Scale of Sup fig 1 is not clear? What exactly is shown?

As indicated by the reviewer we have modified the axis and figure legend to clarify interpretation (now Fig. S3). The scale shows the red fluorescence signal expressed in arbitrary units, as it is being normalized to the average signal obtained from non-infected tissue.

13. It was not clear how data from Fig. 7 was calculated? What are the axis?

We apologized for the lack of detail. Axis and figure legend (now, new Fig. 5B) have been modified, and analysis methodology is included in a separate section of materials and methods (see *Spatial analysis*, P. 29; l. 15) as follows: "For overlap domain analysis, overlap binary masks containing colocalization pixels between glutamatergic afferents and astrocytic binary masks were calculated for each individual slice. Average PRQ images were generated obtaining NAc probability maps in which each pixel value is concealed within 0 and 1 and show the overlap probability in a specific space (Fig. 5B)."

14. Optogenetic parameters should be given in detail, some important parameters are missing

Again we apologize for the incomplete description of this experiment, we have revised the material and method section. The optogenetic parameters used are:

For ChR2: $\lambda = 470 \text{ nm}$ (CoolLED's pE-300white) Intensity curve; 10% (14 mW) – 30% (39.2 mW) – 50% (58.2 mW) - 70 % (70 mW) – 90% (80.7 mW). Optostimulation protocols: (1) 1 ms at 0.2 Hz for Synaptic strength of glutamatergic afferents (Fig. 2B, 3B, 4B) and glutamatergic transmission control CNQX-AP5 (Fig. S4C) or (2) 10 pulses 50 ms at 4Hz - 4 times, 5 s interval for Astrocytic photoconversion + ChR2 optostimulation (Fig. 2E, 3E, 4E, 7, S8, S11D) followed by $\lambda = 405 \text{ nm}$ (Thorlabs M405F1); Light intensity = 1 – 5 mW/cm². Photoconversion protocol: 40 s (Fig. 2E, 3E, 4E, 7, S8, S11D).

For ChrimsonR: $\lambda = 565 \text{ nm}$ (CoolLED's pE-300white system) Intensity curve; 10% (2.56 mW) – 30% (7.2 mW) – 50% (10.9 mW) - 70 % (13 mW) – 90% (14.6 mW). Optostimulation protocol: 1 ms at 0.2 Hz for Synaptic strength of glutamatergic afferents (Fig. 2B, 3B, 4B) and glutamatergic transmission control CNQX-AP5 (Fig. S4C). $\lambda = 590 \text{ nm}$ (Thorlabs M590F3) Light intensity = 1 – 5 mW/cm². Optostimulation protocol TBS: 10 pulses 50 ms at 4Hz - 4 times, 5 s interval for Astrocytic calcium imaging + ChrimsonR optostimulation (Fig. 2C, 3C, 4C, S5, S11B).

To describe these parameters as clear as possible, we described each of them in each experimental approach and also in the new section "**Optogenetic stimulation**" (P. 23. l. 23). "To determine synaptic strength of glutamatergic afferents, optically evoked excitatory postsynaptic currents (EPSCs) were obtained every 5 s with light pulses (0.1 – 1 W /cm²) using CoolLED's pE-300white system through the microscope objective for full-field optostimulation. To enable evoked EPSCs response comparisons, light stimulation strength and pulse duration remained constant for all the recordings (70% LED intensity, 1 ms). Two different wavelengths were used, 470 nm and 565 nm, to stimulate afferents expressing ChR2 or ChrimsonR opsins, respectively. To avoid differences derived from stimulation intensity, light stimulation parameters remained constant for all these recordings (70% intensity, 1 ms).

Dose-response curves (Fig. S4B) were calculated measuring evoked EPSC amplitudes at different light intensities (10, 30, 50, 70 and 90% LED intensity).

To assess astrocytic response, opsin-transfected afferents were full-field activated at the NAc by an optostimulation protocol (10 pulses 50 ms at 4Hz - 4 times, 5 s interval) which remained constant for all experiments to allow for astrocytic Ca²⁺ activity comparison between pathways. Specific parameters of the stimulation intensity for ChrimsonR or ChR2, used respectively to study astrocytic calcium real-time imaging or photoconversion, are specified in the following sections."

."

15. The method for evaluation of anatomical density of projections should be better explained in the material and methods section – from line 492 on on a separate section with more details. How many slices, how many animals? What was used as a control ROI region?

As suggested by the reviewer, we have included an independent section in materials and methods "**Afferent density and opsin transfection.**" (P. 23, l. 5), detailing analysis methodology: *"To measure afferent's density in the NAc, ROIs were delimited manually for AcbC and AcbSh and fluorescence intensity ($\Delta F/F_0$) was calculated for each subregion, being F_0 the background ROI located in a region outside the NAc with no afferent's innervation. Final afferents fluorescence (a.u.) was expressed as the individual AcbC and AcbSh ($\Delta F/F_0$) values relative to the transfection degree ($\Delta F/F_0$) measured at their respective injection site."* Furthermore, number of infections and animals used for each experiment is properly described in the results section of the manuscript and/or supplemental statistics table.

16. In a similar note, the PRQ method should also be described in a separate section

As indicated by the reviewer, we have included the new section **Partition in Regular Quadrants (PRQ)** (P. 27, l. 19) explaining PRQ method, followed by the section **Spatial analysis** (P. 28, l. 11), in which detailed descriptions of spatial analysis conducted in PRQ images are provided.

17. Regarding the same topic as above, authors should also clearly state when ChR2 or Chrimson was used, it is not clear throughout the manuscript.

We have provided specific details both in the manuscript and in materials and methods to indicate which opsin was used as stimulation system for each experiment. Both ChR2 and ChrimsonR were used to electrophysiologically characterize the synaptic strength of the glutamatergic afferents (P, 7; l, 1), demonstrating the possibility to use them equally as stimulation system (Fig. S4). For CaMPARI_{GFAP} calcium imaging experiments, ChrimsonR-tdTom was used as optostimulation system in order to perform simultaneous imaging recordings in the blue light spectrum without interfering with afferents activation (P, 7; l, 14). For CaMPARI_{GFAP} photoconversion experiments, fluorescent reporter of the opsin was changed (EYFP) to avoid crosstalk with CaMPARI_{GFAP} Red signal, using ChR2-EYFP as optostimulation system (P, 8; l, 6).

18. Line 126 remove "medium spiny neurons" because you are evaluating the whole region, not specifically these neurons (for anatomical experiments)

As indicated by the reviewer, in the anatomical experiments we have changed medium spiny neurons (MSNs) to neurons (P. 7, l. 5)

19. The use of GABA A receptor blockers in the recording medium should be properly discussed, especially considering the neuronal constitution of the NAc, which is mostly GABAergic neurons and interneurons.

We thank the reviewer for the important point. Since the NAc is mostly comprised of

GABAergic medium-spiny projection neurons (MSN, > 90%), picrotoxin (50 μ M) was included in the ACSF to hamper the influence of MSN–MSN local synaptic communication as well as inhibitory inputs from fast-spiking GABAergic interneurons within the nucleus accumbens (lateral and feedforward inhibition). This way the activity directly evoked by specific glutamatergic afferents optostimulation was underscored. As pointed out by the reviewers 1 and 3, the following clarifying sentences have been included: “ACSF was supplemented with 0.05 mM picrotoxin ($GABA_A$ receptors antagonist) to eliminate the influence of feedforward/lateral inhibition^{41,74,75}.” (P. 22; l. 15) and (P. 19; l. 22) “The existence of these astrocytic networks was revealed by blocking lateral inhibition and forward inhibition using picrotoxin and preserved when neuronal activity was blocked with the Na^+ channel blocker TTX, highlighting that defined astrocytic ensembles respond to a specific neuronal input. However, of deep interest would be to further study the implications of activating a profile of astrocytes in less-innervated areas in response to glutamatergic stimuli to enlarge the missing links of the physiological picture.”

20. Sup fig 3 -What data supports the use of this method to observe Ca^{2+} spike amplitude? Does it have resolution to do so?

Ca^{2+} spike amplitude parameter is being assessed following same approach as the one used by previous reports (Mederos et al. 2018). In the present work, the signal is analyzed by real-time CaMPARI_{GFAP} green fluorescence signal. Results are detailed in Figure S5 and methodology is included in Materials and Methods Ca^{2+} imaging section (P. 24, l. 13).

21. Line 155-156 – remove opposite; use different

Following reviewer’s indication, we have substituted the terms opposite by different.

22. MPEP experiment: the responses depicted in the graphs refer to core or shell? I assume is shell?

We apologize for the lack of detail, MPEP (Metabotropic glutamate 5 (mGlu5) receptor antagonist; 2C₃, 2E₃, 3C₃, 3E₃, 4C₃ and 4E₃) and dopaminergic antagonists (SCH23390 and Haloperidol; Fig. S11B₂) plotted responses are the result of AcbC and AcbSh combined data. Since no differences were found between subregions (data not shown), data was pooled together. We have corrected and clarified the writing, and we have referenced all the panels of the figures, including this information, in figure legends.

In addition, we have noticed an error regarding the Amyg MPEP data, as the values included in the figure and the subsequent description of the results were misplaced (Fig. 3C₃). Although the statistic does not change after the correction, and therefore the interpretation of the results remains unchanged, we are sincerely sorry.

23. Maybe it is because I am not familiarized with this methodology, but Fig. 2D3 is not clear to me. What do you want to depict? Distance from?

We have included a new section in materials and methods (see *Spatial analysis*, P. 28, l. 11) detailing this information as follows: “To measure fluorescent changes across space, two lines were depicted in the CaMPARI_{GFAP} Red PRQ images starting from dorsal regions and following nucleus anatomy (see as example Fig. 2D-E). Fluorescence quantification (a.u.) within these lines shows signal variation from distance 0 until the complete line’s length; 25 pixels (1250 μ m) for AcbC and 44 pixels (2200 μ m) for AcbSh. Astrocytic activity quantification was determined as the average line value of CaMPARI_{Red} fluorescence (a.u.) in each subregion.”. We have also modified the figure legends.

24. I suggest that the mitochondrial data is provided in a distinct section

As suggested by reviewers 1 and 3, we have reorganized results section, merging former Figures 5 and 7 in a new Figure 5 (and new Fig. S10) and we have separated Figure 6 with the mitochondrial data into an independent section “**Profile of mitochondrial DNA expression and Ca²⁺ activity of AcbC and AcbSh astrocytes**” (P. 15, l. 3).

25. Legends fig 3e3 missing, same in 4, same in 5C1

We apologize for the mistakes. We have carefully revised the present version of the manuscript.

26. Scale in fig4c2 is missing

We apologize for the mistake, Fig. 4C₂ scale has been corrected and heatmap units ($\Delta F/F_0$) have been included in figure legend.

27. Line 301, 325, 422-423, not clear

We have revised the text and clarified the issues in the corresponding text:

Lines 301-423, we have reorganized this results section as “**Profile of mitochondrial DNA expression and Ca²⁺ activity of AcbC and AcbSh astrocytes**” (P. 15, l. 3).

(325) This sentence has been rewritten: “*Next, we explored the differences in astrocytic response between the three glutamatergic inputs, comparing the strength of astrocytic activation among pathways and analyzing the anatomical location of each pattern of astrocytic activity (Fig. 5). As shown in Figure 5A and S5, vHip afferents let to significantly higher fluorescence intensity compared to mPFC (i.e. higher astrocytes Ca²⁺ activity) ($p = 0.024$; Fig. 5A₂) or Amyg ($p = 0.015$; Fig. 5A₂).*”

(422-423) We reviewed the idea: “*The existence of these astrocytic networks was revealed by blocking lateral inhibition and forward inhibition using picrotoxin and preserved when neuronal activity was blocked with the Na⁺ channel blocker TTX, highlighting that defined astrocytic ensembles respond to a specific neuronal input. However, of deep interest would be to further study the implications of activating a profile of astrocytes in less-innervated areas in response to glutamatergic stimuli to enlarge the missing links of the physiological picture.*”

28. Describe data of fig6a and b in results text

As indicated by the reviewer, we describe Fig. 6A and 6B in the results section:

“*To further explore this fact, we monitored real-time CaMPARI_{GFAP} green fluorescence signal in basal conditions and observed that AcbC astrocytes showed higher number of responding ROIs ($p < 0.001$) and calcium spike frequency ($p < 0.001$) (20.6 ± 2.55 % response; 0.12 ± 0.02 Ca²⁺ spike frequency (min^{-1})) when compared to AcbSh astrocytes (9.53 ± 1.15 % response; 0.05 ± 0.006 Ca²⁺ spike frequency (min^{-1})) (2240 ROIs, $n = 36$ slices, 20 mice; Fig. 6A₁₋₃). Same result was obtained when studied by CaMPARI_{GFAP} Red photoconversion, AcbC astrocytes displayed more activity than AcbSh ($p = 0.04$; $n = 20$ slices, 12 mice; Fig. 6A₆).*” (P. 15, l. 8).

“*After astrocyte identification by their electrophysiological properties, showing no electrical excitability when depolarizing current steps were applied (Fig 6B)*” (P. 15, l. 24).

29. Line 441 – co-stimulated responses should be changed to stimulation of 2 or 3 inputs simultaneously ...

Following the reviewer suggestion, we have replaced the terms.

30. The methodology could be better explained; for example it is not clear when authors

present CaMPARI or photoconverted CaMPARI data.

As indicated, we have thoroughly modified the text, materials and methods section and changed axis figures to provide a clearer description regarding this point. We now systematically refer to the expression of CaMPARI_{GFAP} for GECI properties as *real-time imaging of CaMPARI_{GFAP} Green fluorescence* and related to the photoconversion of CaMPARI_{GFAP} as *CaMPARI_{Red}*.

31. the graphs axis are not always clear to interpret what is represented. Maybe add more information to the results section or legends?

We apologize for lack of detail, we have thoroughly revised the manuscript text, adding more information about the methodology and including changes in axis and figure legends (See comment #7) to clarify data interpretation.

32. P values of Pearson correlations are missing in legends

We have carefully revised all legends and corrected these mistakes.

33. Line 352-354 – I think is the other way around?

We have included a remark to clarify the writing regarding this point: “*Interestingly, after comparison of % overlap area of the different afferents we found characteristic intrinsic features depending on the glutamatergic or dopaminergic transmission ($p < 0.001$, Fig. 5C). Overall present results show distinct astrocytic calcium activation in response to principal glutamatergic nuclei and define spatially segregated regions enclosing direct interaction between both circuit elements, which are only found in the glutamatergic system.*” (P. 14, l. 22).

Personal notes:

- I found really difficult to follow figures having E1, E2, D1, D2, D3, D4 etc, I suggest using exclusive letters for each section of figures

- The mit data is interesting but, in my modest opinion, as it is, does not add much to the main findings.

We thank the reviewer for the suggestions, and we appreciate the effort in helping us to clarify the issues.

Reviewer #2:

Investigation of pathway-specific activation of glutamatergic fibers to the nucleus accumbens (NAc) placed the distinct neuronal input and/or output features in the centre. Therefore, conjuring up the role for distinguishable astrocyte assemblies integrating pathway-specific glutamatergic transmission is certainly an important and sensible approach [Kardos et al., Molecular plasticity of the nucleus accumbens revisited - astrocytic waves shall rise; Mol. Neurobiol. 2019, 56, 7950-7965; doi: 10.1007/s12035-019-1641-z]. In order to study functional neuron-astrocyte circuitries in the NAc, Authors devised a fluorescent technique based on calcium-modulated photoactivatable ratiometric integrator (CaMPARI) that undergoes irreversible green-to-red fluorescence conversion upon coincident elevated intracellular Ca(2+) and ultraviolet light illumination [Fosque et al., Neural circuits. Labeling of active neural circuits in vivo with designed calcium integrators; Science 2015, 347, 755-760; doi: 10.1126/science.1260922]. Authors claim that the selective optostimulation of main glutamatergic inputs (i.e. prefrontal cortex, basolateral amygdala and ventral

hippocampus) induces astrocytic Ca(2+) activities mediated by metabotropic glutamate receptor mGluR5 that do not coincide with glutamatergic innervations, suggesting unexpected neuron-astrocyte circuitries. Interestingly, the differences in basal Ca(2+) dynamics between the NAc shell and core astrocytes were associated with differences in mitochondrial DNA copy number, exhibiting molecular heterogeneity in the regulation of their mitochondrial genomes.

We thank the reviewer for the supportive comments on the context of the tools used and the helpful suggestions that allowed the clarification of important issues.

Unfortunately, the description of the novel approach applied to monitor astrocyte activity (CaMPARIGFAP) is not adequately addressed at present. My specific questions (Qs) are as follows:

1. Why should the elevation of intracellular Ca(2+) by ATP decrease the CaMPARIGFAP fluorescence in astrocytes? 2. Why does the ATP-induced decrease in CaMPARIGFAP fluorescence demonstrate “the ability of the molecule to monitor Ca(2+) dynamics”?

The confusion of the reviewer was certainly due to the lack of clarity of the previous version in relation to our original presentation of the CaMPARI_{GFAP} tool. In the current manuscript, we have performed a new set of experiments, expanded the explanation regarding the experimental design, and substantially revised the description part of our experimental approach, highlighting the description of the CaMPARI_{GFAP} procedure. I) In order to clarify all the technical review concerns regarding CaMPARI_{GFAP}, we have included schematic representations (Fig. 1C₁ and 1D₁) with respect to CaMPARI_{GFAP} experimental approaches. II) Several laboratories have reported the ATP-evoked increases in intracellular calcium in astrocytes (Salter and Hicks, 1994; Centeneri et al., 1997; Bowser and Khakh, 2004; Newman 2005; Perea and Araque 2005; Beierlein and Regehr, 2006; Piet, and Jahr, 2007; Shigetomi et al., 2010, Molnár et al., 2011; Navarrete et al., 2012; Guerra Gomes et al., 2018). We have now directly evaluated the effectiveness of our BAPTA-loading method to quench Ca²⁺ signals in astrocytes, as requested by the reviewers (1 to 3), and Thapsigargin (1 μM), which depletes the intracellular Ca²⁺ stores by inhibiting the Ca²⁺ ATPase (Navarrete and Araque, 2008, Perea and Araque, 2005). These experiments have been included in the text and new Figures to clarify the issue (See also comment #1 to Referee 1).

“Afterward, we studied both the ability to track real-time Ca²⁺ astrocytic activity and photoconversion properties²⁷. Using local application of ATP (20 mM) through a micropipette, which reliably elevates intracellular Ca²⁺ levels in NAc astrocytes (see²⁹), we showed a transient decrease in CaMPARI_{GFAP} green fluorescence in astrocytes (Fig. 1C₁₋₃)^{30,31}. This change in the green fluorescence signal was prevented by perfusing 1 μM thapsigargin (1.67 ± 1.67 % in thapsigargin vs. 98.9 ± 1.11 % in control; n = 6 and n = 9 slices, in thapsigargin and control conditions respectively; p < 0.001) which depletes the internal stores by inhibiting the Ca²⁺ ATPase. The same was achieved by loading BAPTA (20 mM) into the astrocytes syncytium using a patch pipette (p < 0.001; Fig. 1C, see Fig. S1 for a representative example of astrocyte intracellular loading with BAPTA and biocytin, followed by streptavidin-Alexa 647 staining). It is well known that BAPTA spreads via gap-junctions into the astrocyte syncytium network, blocking astrocytic Ca²⁺ signaling throughout the slice³²⁻³⁷. These results demonstrate the viability of the molecule to monitor Ca²⁺ dynamics in astrocytes.

In parallel, application of 405 nm light during a fixed temporal window (10 s) right after ATP local delivery led to green-to-red photoconversion of activated astrocytes (Fig. 1D); to note, no tissue damage was detected due to 405 nm light application³⁸⁻⁴⁰ (Fig. S2, see Materials and Methods). To cover large areas of tissue, CaMPARI_{GFAP} Green and Red fluorescences were

measured post-hoc after PFA fixation. As shown in Figure 1D the CaMPARI_{GFAP} Red and Green fluorescences changed according to the distance from ATP application ($\Delta F/F_0$, Fig. 1D₂). These fluorescence changes were also analyzed in presence of thapsigargin or BAPTA infusion, revealing no photoconversion nearby the stimulus (Fig. 1D), which shows that the green-to-red photoconversion ratio correlated with Ca²⁺ activity (red fluorescence is increased at closer distances to the micropipette while green fluorescence is decreased).” (P. 5;l. 6).

We have also clarified these issues in the Experimental Procedure section (P. 21;l. 1).

3. Authors claim the application of UV light during a fixed temporal window, however, they do not explicate why. In reality, UV light may seriously affect cellular viability conjuring up the question how does the fixed temporal window ensure that this would not be happening under the explicit application protocol?

We thank the reviewer for pointing out this issue, and we apologize for the lack of explanation. In line with comment #1, we have included the following sentence to better explain CaMPARI_{GFAP}'s procedure (P. 26 l. 16): *"light stimulation protocols were applied: (1) optogenetic stimulation protocol followed by photoconversion protocol (40 s of violet light ($\lambda = 405 \text{ nm}$)) or (2) basal condition, in which only photoconversion protocol was applied. Full-field ChR2 optogenetic stimulation was delivered using CoolLED's pE-300white system through the microscope at a fixed 70% light intensity ($\lambda = 470 \text{ nm}$; 80 mW/cm^2) using a 10x microscope objective. Violet light (405 nm) was delivered obliquely at a fixed intensity (1 - 5 mW/cm^2 (Thorlabs M405F1))."*

To determine this temporal window, we took into consideration the astrocytic response we meant to capture and the temporal resolution of the experiment. We also accounted for the dose of 405nm light delivered to the tissue (0.2 J/cm^2) under these conditions, being below the cytotoxic levels ($\sim 50 \text{ J/cm}^2$) reported for mammalian cells (Ramakrishnan et al. 2016, Wäldchen et al. 2015 and Ramakrishnan et al. 2014). (P. 30; l. 4).

Furthermore, being this point critical for the present study we have performed a battery of experiments to evaluate tissue damage due to 40 s 405 nm light delivery (new Fig. S2). We have analyzed the violet illumination effects in the circuit excitability by electrophysiological recordings of NAc neurons before and after 405 nm light treatment without observing differences in the neural viability (Fig. S2B₁₋₂). Neither did we find changes in astrocytic calcium dynamics (Fig. S2C), which were monitored by real-time CaMPARI_{GFAP} green fluorescence. Moreover, we have performed an immunohistochemistry analysis (Fig. S2D₁₋₂) of Iba1, S100 β , and GFAP markers to compare microglia and astrocytes reactivity in control slices and slices treated with our 40 s violet light photoconversion protocol. Astrocytic reactivity was assessed not only by the astrocytic marker S100, but also by the GFAP marker whose expression is enhanced in reactive astrocytes (Sofroniew and Vinters, 2010). We did not detect any difference between control and slices treated with 405 nm light regarding the labeled area of every marker used (Fig. S2D₂), indicating no increase in the size or ramification of the cells typical of gliosis (Sofroniew and Vinters, 2010; Ahmed et al., 2007). Overall, present results in line with previous reports indicate (i.e. Ramakrishnan et al., 2016; Ramakrishnan et al., 2014; Wäldchen et al., 2015) that the 405 nm light dose used in the present study does not affect the cellular viability of the samples.

4. How do we know that the green-to-red photoconversion occurred in those astrocytes that were active at the moment of illumination if ATP stimulation decreases the CaMPARI_{GFAP} fluorescence (c.f. Q1 and Q2)?

This concern was directly addressed in presence of Thapsigargin or BAPTA in astrocytes. By monitoring in real-time the green fluorescence of CaMPARI_{GFAP} we have observed a

fluorescence decrease that moves away from the pipette where we locally applied ATP. The same experimental approach was used when exposing the slice to 10 s of ultraviolet light at the same time as the local application of ATP (to achieve red-to-green photoconversion). As shown in *new Figure 1*, this wave of fluorescence change (either green or red) does not occur when blocking calcium ATPase throughout the slice (using thapsigargin), or when specifically chelating calcium in the astrocyte network (experimental approach with BAPTA). This result shows that the green-to-red photoconversion degree correlated with Ca^{2+} activity; red fluorescence is increased at closer distances to the micropipette while green fluorescence is decreased. (P. 5;l. 7).

Furthermore, we have included a more detailed CaMPARI_{GFAP} description in Results section (P. 4;l. 18): “CaMPARI^{26,27}, a genetically encoded Ca^{2+} indicator (GECI) expressed in astrocytes that undergoes irreversible green-to-red fluorescence conversion upon coincident elevated intracellular Ca^{2+} and violet ($\lambda = 405 \text{ nm}$) light illumination²⁷. The imaging of CaMPARI Green fluorescence allows real-time monitoring of astrocytic Ca^{2+} dynamics, while its irreversible photoconversion properties (red fluorescence) enable large-scale spatial analysis of astrocytic activation with precise temporal resolution²⁶⁻²⁸.”

5. My understanding is that the Red/Green fluorescence ratio (FRed/FGreen) of CaMPARIGFAP changes according to the distance from the site of ATP application. The application distance, however, may vary from one experiment to another. I assume that this kind of variability does impacts the quantification of Ca(2+) transients as well. Furthermore, in order to get unconditional data, the application of the CaMPARIGFAP tool necessitates devising relative data such as for example (FACTUAL-FBASAL)/FBASAL.

Besides former Figure 1, photoconversion analysis reported in the present work (astrocytic response to the glutamatergic afferents) is being executed by measuring the relative $\Delta\text{F}/\text{F}_0$ CaMPARI_{GFAP} Red signal, both for basal and optostimulated slices (see materials a methods CaMPARI_{GFAP} photoconversion, Partition in Regular Quadrants (PRQ) and Spatial analysis sections). To unify the way the photoconverted signal is measured across experiments, and following the reviewer's suggestion, we have changed the analysis of new Figure 1 presenting photoconversion data as fluorescence $\Delta\text{F}/\text{F}_0$ (Figure 1D₃). Note that although the intensity of the red fluorescence signal is decreased compared to the green due to PFA fixation (Fosque et al. 2015), the red signal-to-noise ratio does not compromise measurements in astrocytes.

6. The applied confocal microscopy may suffer from out-of-focus light contaminating the region of interest, and thus accurate green fluorescence measurements are not possible in the presence of non-specific background fluorescence, interfering the generation of an accurate FRed/FGreen ratio.

As indicated by the reviewer, all results have been analyzed as relative data (FACTUAL-FBASAL)/FBASAL), considering CaMPARI_{GFAP} Red fluorescence (FACTUAL) at AcbC and AcbSh and a background ROI (FBASAL) outside the NAc (see *Partition in Regular Quadrants (PRQ)* section at materials and methods).

7. As with any activity reporter, a critical aspect of interpreting experiments using CaMPARI is to carefully calibrate readout versus the underlying phenomenon under study; importantly, such calibrations should take place in the preparation of interest (when possible), as factors such as expression level, long-term stability, and light delivery and collection can vary widely. [c.f. Zolnik et al., All-optical functional synaptic connectivity mapping in acute brain slices using the calcium integrator CaMPARI. J. Physiol. 2017, 595, 1465-1477; doi: 10.1113/JP273116]

We thank the reviewer for the critical point. Following the reviewers (1 to 3) indications we have included new panels with the plots of transfection degree (*new Fig. S12*). Moreover, the astrocytic activity measurements have been normalized to their basal control, expressed as change from basal, to minimize external factors' variability. For calcium imaging experiments, basal activity recordings were performed in the first place and, afterward, the same ROIs were analyzed in response to afferents optostimulation (see Ca²⁺ imaging section at materials and methods; P.24, l. 13). For photoconversion experiments, consecutive slices within the NAc (containing the same levels of afferent's opsin and CaMPARI_{GFAP} transfection) were paired into basal and optostimulated conditions, undergoing in parallel through all experimental steps including PFA fixation and confocal image collection (see CaMPARI_{GFAP} photoconversion section at materials and methods; P. 26, l. 13). Furthermore, light delivery parameters were maintained constant in all experiments for both, afferents optostimulation and CaMPARI_{GFAP} photoconversion. Since photoconversion is linearly dependent on the violet light intensity, all slices were placed at the same distance and orientation to the light beam in order to avoid variation in the scattering pattern.

We apologize for the lack of detail regarding methodology; we have included all this information in the revised materials and methods.

Reviewer #3:

In this manuscript, Serra and colleagues dissected the functional connectivity between distinct glutamatergic neuronal circuits and astrocyte networks in the nucleus accumbens (NAc). To study the neuron-astrocyte interaction in NAc, the authors 'smartly' combined optogenetic stimulation of neurons with the optical monitoring of astrocyte-network activity using genetically encoded Ca²⁺ activity integrator (CaMPARI). The authors used recombinant AAVs to express channelrhodopsin2 (Chr2) in the neurons in various brain areas differentially projecting to the NAc core (AcbC) and shell (AcbSh) region of NAc. These brain regions include the medial prefrontal cortex (mPFC), amygdala, ventral hippocampus (vHip), and ventral tegmental area (VTA). By simultaneously activating neuronal Chr2 (with blue light) and astrocytic CaMPARI (UV-light), the authors tried to capture the astrocyte network activated in response to neuronal activation. As proof of principle, the authors combined

optogenetics and electrophysiology to show a strong positive correlation between glutamatergic fiber innervation and medium spiny neurons (MSNs) responses in the core and shell of NAc. On the contrary, astrocyte network activation (i.e., intracellular Ca²⁺ increase) doesn't correlate with the glutamatergic fiber innervation and neuronal activity patterns. Although the inverse relationship between neurons activation and astrocyte network activity is fascinating, the authors don't provide any potential mechanism driving the differential response of astrocytes on activation of individual glutamatergic pathways. In the current state, the manuscript requires a significant body of work for conveniently establishing the significance of these observations.

We thank the reviewer for the comments that “the inverse relationship between neurons activation and astrocyte network activity is fascinating” and, for the supportive comments on the context and significance of this study and for the helpful suggestions that have improved the manuscript and strengthened the conclusions. We have rearranged and strengthened the ideas exposed in the discussion section of the manuscript.

Major concerns:

1. The authors perform all their CaMPARI experiments in the presence of picrotoxin, quite likely to block the action of inhibitory neurons. Although this is critical information, the authors don't explicitly mention this in the result section and discuss the rationale behind using picrotoxin. Why do they need to block inhibition in their brain slices? This can already induce neuronal activation in the absence of any optogenetic stimulation.

This is a good point that was also raised by reviewer 1. Since the NAc is mostly comprised of GABAergic medium-spiny projection neurons (MSN, > 90%), picrotoxin (50 μ M) was included in the ACSF to hamper the influence of MSN–MSN local synaptic communication as well as inhibitory inputs from fast-spiking GABAergic interneurons within the nucleus accumbens (lateral and feedforward inhibition). This way the activity directly evoked by specific glutamatergic afferents optostimulation was underscored. As pointed out by the reviewer, we have included the following sentences: “*ACSF was supplemented with 0.05 mM picrotoxin (GABA_A receptors antagonist) to eliminate the influence of feedforward/lateral inhibition^{41,74,75}.*” (P. 22; l. 15) and (P. 19; l. 9) “*The existence of these astrocytic networks was revealed by blocking lateral inhibition and forward inhibition using picrotoxin and preserved when neuronal activity was blocked with the Na⁺ channel blocker TTX, highlighting that defined astrocytic ensembles respond to a specific neuronal input. However, of deep interest would be to further study the implications of activating a profile of astrocytes in less-innervated areas in response to glutamatergic stimuli to enlarge the missing links of the physiological picture*”.

2. The authors show that astrocytes Ca²⁺ increase is more robust in regions with less glutamatergic fiber innervation (see Fig 2. B3/C2 and Fig 4. B3/C2) and is dependent on mGluR5 activation. If NAc astrocytes activation is dependent on glutamate release, what is the source of this glutamate? If the source is synaptic glutamate release, we should expect an overlap between glutamatergic neuronal innervation and astrocyte activation.

We thank the reviewer for this relevant remark since initially we also expected overlap between glutamatergic innervation and astrocytic response. We were surprised to observe that this was not the case since the lack of correlation between afferents and astrocytic activity was consistent when assessed from two different experimental approaches (e.g. Fig. 2A-C and 2D-E), concluding in both cases that astrocytes do not respond exclusively to glutamatergic inputs in highly innervated areas. Nevertheless, there are restricted regions in the NAc in which astrocytes are responding in dense glutamatergic afferents innervation regions, which we defined as overlap areas (as an example, Fig. 2F). We reason that since the main source of glutamate is being released at these innervated regions (as shown by the correlation of EPSCs amplitude and afferent's density, for example in Fig. 2B2), initial astrocytic response should be directly triggered by mGluR5 from these overlapping areas. Our hypothesis is that an astrocytic network is responding to each specific pathway activation, broadening the calcium response further into the NAc.

It could be discussed if astrocytic calcium activity reported in low innervated areas is due to astrocytic network intrinsic properties as we propose, or if it is due to indirect activation of local NAc neurons not silenced by picrotoxin. To further explore this possibility, we have repeated the experiments in presence of TTX 1 μ M (*new Fig. S8*), blocking neural activation in the NAc. We showed no difference in the astrocytic response to glutamatergic afferents when applying TTX (*new Fig. S8D and S8E*), further suggesting activation of an astrocytic network.

We have included the following sentences in the discussion section: “*The existence of these astrocytic networks was revealed by blocking lateral inhibition and forward inhibition using picrotoxin and preserved when neuronal activity was blocked with the Na⁺ channel blocker*

TTX, highlighting that defined astrocytic ensembles respond to a specific neuronal input. However, of deep interest would be to further study the implications of activating a profile of astrocytes in less-innervated areas in response to glutamatergic stimuli to enlarge the missing links of the physiological picture.” (P. 19, l. 9).

3. From Fig. 2C1 (and 2E2), it seems AAV-based expression of CaMPARI in non-homogenous across NAc, i.e., more astrocytes express CaMPARI in shell than the core. Hence, the difference in the astrocyte activity and the innervation profile of afferents from various brain regions probed in this study can simply emerge from the number of astrocytes expressing CaMPARI in shell vs. core.

This is indeed an interesting observation that had not escaped our attention. We have measured CaMPARI_{GFAP} AAV transfection at NAc in all experiments (See *new Fig. S12*) and although we account for certain variability, no differences in expression levels are found between AcbC and AcbSh or between experimental conditions (*new Fig. S12A₂ and S12B₂*).

4. In Figure 4, there is a discrepancy between C2 and E2. Based on the images shown, there is a reasonably strong activation of AcbSh astrocytes (E2), but in C2, this seems to be relatively mild. Also, when vHip afferents are optogenetically stimulated, astrocytes in the entire area covered by these afferents show a Ca²⁺ increase (Fig. 4F1). At least, in this region, the reverse correlation between neuronal and astrocytic activity is not as evident as seen for mPFC and Amygdala (compare figures 2F1, 3F1, and 4F1).

We agree with the reviewer, despite the activity profiles between subregions is maintained in both experiments (more activity is registered in AcbC with respect to AcbSh), there was a discrepancy in the magnitude of the astrocytic response between calcium imaging results (previous 4C) and photoconversion experiments (4E). Since the number of vHip samples ‘n’ in calcium imaging experiments was reduced compared to the other glutamatergic nuclei, we have now increased the ‘n’ until the sample number as presented for mPFC (n = 25 slices, 8 mice). After this increase, we confirm strong activation of AcbSh astrocytes, disappearing the disparity between both experimental approaches (4C and 4E), which was probably due to a poorer representation of the effect in the initial pool of data.

We are also in agreement with the second remark highlighted. As stated in the manuscript (P, 12; l, 14) when vHip pathway is assessed, we observed spatial overlap between dense innervation and astrocytic response in the AcbSh that was not present in AcbC (*Fig. 4G*), which contrast with the results from mPFC and Amyg (*Fig. 2G and 3G*). Nevertheless, total NAc spatial overlap remains significantly smaller than the astrocytic activation area (*Fig. 4F₂*) also indicating activation in low afferent regions as for the other inputs (*Fig. 2F₂ and 3F₂*). Although we cannot provide the biological significance behind this phenomenon, present results suggest different astrocytic processing in response to the glutamatergic pathways. They could also be resolved as distinct astrocytic processing between NAc subregions. To further explore both ideas later in the manuscript we compare the astrocytic response between glutamatergic pathways (*new Fig. 5*), and we perform experiments to survey for astrocytic heterogeneity between subregions (*new Fig. 6*).

5. In general, AcbSh astrocytes always respond to optogenetic stimulation of glutamatergic afferents (see figures 2-4 E2), and when afferents from vHip are stimulated, then astrocytes in AcbC also strongly responded. Hence, from this point of view, it will be difficult to conclude that there is any specific co-relation between glutamatergic afferents innervation and astrocyte activity.

We apologize for the lack of clearness in the conclusions presented in the manuscript, which has been fully revised and corrected. We did not report any correlation per se between

innervation and astrocytic response.

As stated by the reviewer, from figures 2E, 3E, and 4E, we conclude that astrocytes respond in all cases to individual pathway stimulation. By analyzing the calcium activity individually, we cannot demonstrate if this increase in activity is due to an unspecific response to glutamate or if astrocytes show synapse-specificity. This is why we have performed an additional analysis (Fig. 5A₁₋₂) in which we test differences between the astrocytic responses triggered by each glutamatergic pathway. In the previous version of the manuscript, we considered AcbC and AcbSh separately for this analysis (former Fig. 5). To better survey for differential response between pathways, we have changed the statistical test by running a Two-way ANOVA in which we analyze together the entire NAc astrocytic response for each input (Fig. 5CA₂). As can be observed, in agreement with our initial conclusions, their astrocytic activity in response to vHip is increased with respect to the other pathways, suggesting synapse-specificity.

6. How does the variability in the mitochondrial DNA (mtDNA) copy number (Fig. 6B1, B2) relate to the differential astrocytic Ca²⁺ response seen between AcbSh and AcbC (Fig. 6B2)? What is the source of a large scatter in the mtDNA/cell in AcbC (Fig. 6B2)? Most of the cells have similar mtDNA copies in AcbC and AcbSh (Fig. 6B2); few AcbC cells show a considerable variation in the mtDNA copy number. In short, the key question is, how does differential mitochondrial DNA copy number in astrocytes allow for differential input integration of glutamatergic pathways by astrocytes?

This key point is currently an important open question in the field. The role of mitochondria as a local source of calcium in astrocytes, in ATP production, and glutamate metabolism is well-documented (Jackson and Robinson 2018, Agarwal et al. 2017). As stated for Reviewer 1 comment #8, Our results showed enrichment of mitochondrial DNA copies in a fraction of the AcbC astrocytes collected for analysis (Fig. 6B), which is consistent with the distinct Ca²⁺ signaling observed between NAc subregions (Fig. 6A). These results are in line with recent single-cell studies which explore the transcriptome in situ (Batiuk et al. 2020, Bayraktar et al. 2020, Ohlig et al. 2022), which find differential molecular and Ca²⁺ signaling between astrocytic subtypes (Batiuk et al. 2020) and also show the region-restricted mapping of genes related to mitochondrial functions, prompting the idea that metabolic specialization may be more region-specific depending on local neuronal networks (Ohlig et al. 2022). Unlike these massive sequencing techniques, our patch-dPCR provides accurate quantification of absolute mitochondrial DNA copies at a single-cell resolution which we believe is an important feature that compensates for the restricted population of cells collected due to manual sampling. Although future work is needed to further dissect molecular heterogeneity in the NAc astrocytes, our results of differential Ca²⁺ signaling in combination with mitochondrial DNA copies provide an unprecedented characterization of the nucleus in line with recent evidence reported in the field. We have included the following sentence to clarify the issue: “*These results, in concordance with transcriptome in situ studies⁵⁴⁻⁵⁶, indicate an increased Ca²⁺ activity coupled with a mtDNA copy number enrichment in astrocytes. This is indeed an exciting issue, giving rise to the idea that metabolic specialization is region-specific and depend on local neuron-astrocyte circuits. However, the elucidation of the mechanisms underlying this phenomenon will likely engage extensive new research in the future.*” (P. 16, l. 7)

7. Figure 7 is entirely confusing, and it is unclear what message the authors want to convey. The plots presented in this figure are non-intuitive and show contrary information to those shown before in Fig. 2, 3, and 4. Also, there is some level of redundancy between figures 5 and 7.

Following reviewer's indications (see Reviewer 1 comment # 24), we have restructured the manuscript unifying previous Figures 5 and 7 and leaving in a separate section Figure 6 for better data comprehension. Under this new arrangement, present Figure 5 summarize the results exploring differences in the astrocytic response triggered by the different inputs coming to the NAC. Figure 5A, compares the astrocytic calcium response per se, while Figure 5B (former Fig. 7A) focuses on the differences between astrocytes and afferents interaction (overlap domains) revealing distinct neuronal-astrocytes circuitries for each glutamatergic pathway.

Furthermore, in agreement with the reviewer, to avoid misconceptions regarding redundancy we have included former Figure 5B as a supplementary figure (Fig. S10), including a more detailed methodology description in materials and methods (See Spatial analysis section; P. 28, l. 11). Following the same line, we have amended all figure axis regarding these results (see Reviewer 1 comments #7 and #13).

8. What is the mechanism by which the activation of the amygdala can suppress the activation by vHip (Fig. 8B1, D1), given that mPFC doesn't suppress the activation of vHip (Fig. 8C1)? Why would co-stimulation of all pathways (mPFC, amygdala, and vHip) don't induce Ca²⁺ transients (Fig. 8D1-D3), given that individual pathways activation-induced Ca²⁺ transients in AcbSh and AcbC? The authors don't provide any mechanism behind this crucial observation.

We have discussed the issue by adding the following paragraph (P. 19, l. 23):

Our results indicate that the sum of glutamatergic inputs decreases astrocytic activity, indicating the synaptic information processing by astrocytes in the nucleus accumbens. "These results agree with the reported calcium activity regulation by different synaptic inputs in the hippocampus²⁵. While further studies, out of the scope of the present work, are required to elucidate the underlying molecular mechanisms of this phenomenon, it can be hypothesized that it might be due to the interaction of the intracellular signaling pathways stimulated by both synaptic inputs (see^{71,72}). This lack of linearity shows that the integration property orchestrated by astrocytes in the NAc (e.g., the ability to increase the signal-to-noise ratio (see⁷³)), could mechanistically explain the divergent physiological and behavioral responses produced by the activation of different glutamatergic inputs to the NAc, revealing the neuron-astrocyte network as a critical center for the integration properties of the NAc^{9,22,23,41,42,69,70}."

Minor concerns:

- *Line 887 – Fig. 2D4 is related to the neuronal afferents and not the astrocyte CaMPARI signal.*

We apologize for the mistakes. We have carefully revised the present version of the manuscript.

- *What does $\Delta F/F_0$ represent in Fig. A2? There is no concept of time in this image.*

In line with reviewer 1 comment # 15, a separate section on "**Afferent density and opsin transfection.**" has been included (P. 22, l. 17), detailing the present analysis methodology as follows: "To measure afferent's density in the NAc, ROIs were delimited manually for AcbC and AcbSh and fluorescence intensity ($\Delta F/F_0$) was calculated for each subregion, being F_0 the background ROI located in a region outside the NAc with no afferent's innervation. Final afferents fluorescence (a.u.) was expressed as the individual AcbC and AcbSh ($\Delta F/F_0$) values relative to the transfection degree ($\Delta F/F_0$) measured at their respective injection site."

F_0 does not account for time, but as a FBASAL reference in order to get unconditional data (see Reviewer 2, comment #3), avoiding unspecific tissue fluorescence contribution.

• *Typo: the last section should be Fig. 8, but Fig. 7 is mentioned throughout the text.*

We apologize for the mistakes. We have carefully revised the present version of the manuscript.

We thank the reviewer for the helpful comments.

REVIEWERS' COMMENTS

Reviewer #2 (Remarks to the Author):

Authir's answers can be accepted.

Reviewer #4 (Remarks to the Author):

This is just to state that I join the review process only now (it's the first time I see the paper). A very thorough paper from the Navarrete group, showing the response to the glutamatergic inputs to the NAc (separately from mPFC, Amyg, vHip, and together). They take advantage of the CaMPRI tool (and it's ability to monitor Ca as well), and collected a huge amount of data.

Major:

Figure 7 brings up an interesting pattern that the writers choose to ignore: I think the only thing in the combined experiment that is clearly seen, is that the amygdala has an inhibitory effect on every other pathway it is combined with. I would mention it in the result, and discuss it in the discussion.

Figure 2F, 3F & 4f – the paper discusses astrocytes activity in response to glutamate afferents. However, they mostly don't fit... It's discussed, but not enough.

Minor:

- 1) The C1, C2, C3 (C is just an example) is terribly confusing. I would switch it to normal letters.
- 2) In Figure 1 C1 – I do not understand the relation between the colors of triangle from low to high Ca activity and the colors of the astro above it. They go in opposite direction. (I do understand what you mean, I just find it confusing :)
- 3) P13, row 14-18 "...supporting the idea that NAc astrocytes can discern the origin of each of the glutamatergic inputs". That's going a bit far, no? It might be the astrocytes responding to the neural reaction to the glutamatergic inputs. Your method is precise in time, so I wouldn't make such strong statements.
- 4) The mitochondria don't fit the general glutamatergic story, it's just stuck in the middle... As far as I'm concerned it can be removed from the paper.

(Reviewer comments are in *blue italics*; sentences included in the manuscript are in *italics*).

Reviewer #4

This is just to state that I join the review process only now (it's the first time I see the paper). A very thorough paper from the Navarrete group, showing the response to the glutamatergic inputs to the NAc (separately from mPFC, Amyg, vHip, and together). They take advantage of the CaMPRI tool (and it's ability to monitor Ca as well), and collected a huge amount of data.

We thank the reviewer for considering the manuscript.

Major:

Figure 7 brings up an interesting pattern that the writers choose to ignore: I think the only thing in the combined experiment that is clearly seen, is that the amygdala has an inhibitory effect on every other pathway it is combined with. I would mention it in the result, and discuss it in the discussion.

We thank the reviewer to raise this relevant question. We had not introduced this idea originally in the manuscript because of its speculative nature, but we are happy to do it now in the revised manuscript. We have introduced “*Strikingly, stimulation of the amygdaloid afferents induced an inhibitory effect on astrocyte responses regardless of the pathway with which it is co-stimulated (Fig. 9A,D and J). Given these data, it would appear the Amyg has a dominant influence over astrocyte circuitries.*” in results section P. 16 L. 10 and “*In a manner that depends on the transient state of the system, dopamine plays a complex role in the gating of afferent input to the (Floresco et al., 2001; Howland, Taepavarapruk and Phillips, 2002). According to the data obtained in Figure 9, there is the possibility that the afferent activity from the Amyg can facilitate the release of mesoaccumbens dopamine efflux in the NAc (Howland, Taepavarapruk and Phillips, 2002; Bercovici et al., 2018). This co-release of glutamate and dopamine could activate a number of different cellular mechanisms that would lead to an inhibition of Amyg-astrocyte circuitries.*” in discussion section P. 21 L. 1

Figure 2F, 3F & 4f – the paper discusses astrocytes activity in response to glutamate afferents. However, they mostly don't fit... It's discussed, but not enough.

The reviewer referred figures have been re-organized and re-named in the last version of the manuscript to Fig.3E, 5E, 7E. As requested by the reviewer, we have re-organized and extended our discussion regarding this topic as follows:

“We found that NAc astrocytes respond to excitatory inputs in a pathway-specific way, since optostimulation of each glutamatergic input results in different profiles of high astrocytic Ca²⁺ activity within the NAc (Fig. 8A-B). Astrocytes did not appear to respond solely in highly innervated regions (Fig. 3E, 5E and 7E); mPFC inputs triggered high Ca²⁺ activity in astrocytes within both NAc subregions (Fig. 3 and 8), while astrocytes highly responsive to the Amyg seemed to be preferentially concealed within dorsal regions of the NAc (Fig. 5 and 8). Further, vHip activated a broad astrocyte network covering most of the NAc (Figs. 7 and 8). The fact that the

astrocytic response is not concealed within the major glutamatergic afferents regions immediately suggests an underlying complex pattern of network activation. This is a characteristic of glutamatergic pathways that is not present within dopaminergic signaling (Fig. 8F and S12). The existence of these astrocytic networks was revealed by blocking lateral inhibition and forward inhibition using picrotoxin and preserved when neuronal activity was blocked with the Na⁺ channel blocker TTX, highlighting that defined astrocytic ensembles respond to a specific neuronal input. However, of deep interest would be to further study the implications of activating a profile of astrocytes in less-innervated areas in response to glutamatergic stimuli to enlarge the missing links of the physiological picture. Moreover, we found spatial differences in the way astrocytes interact with high-density afferent areas among the pathways (Fig. 8C-F). Considering the positive correlation between neuronal response and glutamatergic afferents stimulation, it is more likely that these high-density regions constitute the major source of glutamate release in the NAc. Interestingly, the astrocytic response in these areas rich in glutamate is not only restricted, but also segregated in the NAc space for each individual pathway, revealing input-specific overlap regions. This show the spatial distribution of different neuron-astrocytes circuitries in the NAc, pointing towards the existence of specific hotspots in the glutamatergic system which are not observed in response to the VTA. Further functional dissection of neuro-glia circuits in other brain regions and in response to other neurotransmitters would be of the outmost interest to bring a deeper understanding and to contrast these hypotheses." in discussion section P.17 L.20.

Minor:

1) The C1, C2, C3 (C is just an example) is terribly confusing. I would switch it to normal letters. Changed as indicated by the reviewer.

2) In Figure 1 C1 – I do not understand the relation between the colors of triangle from low to high Ca activity and the colors of the astro above it. They go in opposite direction. (I do understand what you mean, I just find it confusing :)

To clarify the point raised by the reviewer in figure 1C, we have changed from black to white the high Ca²⁺ activity triangle's color in order to match the colors of the astrocytes above.

3) P13, row 14-18 "...supporting the idea that NAc astrocytes can discern the origin of each of the glutamatergic inputs". That's going a bit far, no? It might be the astrocytes responding to the neural reaction to the glutamatergic inputs. Your method is precise in time, so I wouldn't make such strong statements.

As indicated by the reviewer we have modified this sentence "...supporting the hypothesis that NAc astrocytes could discern the origin of each of the glutamatergic inputs." P. 13 L. 18.

4) The mitochondria don't fit the general glutamatergic story, it's just stuck in the middle... As far as I'm concerned it can be removed from the paper.

In agreement with the referee on this point, the main topic of the manuscript is the integration of different glutamatergic pathways to the nucleus accumbens, and that this integration is linked to region-specific and functionally different astrocytes. Following the reviewer's suggestion, the mitochondrial data have been removed.

Nonetheless, the finding that calcium dynamics in astrocytes is associated with mitochondrial DNA copy number indicates a molecular mechanism that underlies the functional heterogeneity of astrocytes. The difference in single-cell astrocyte mtDNA copy number, a tightly controlled and regulated cell process, provides initial evidence to hypothesize that there is a region-specific metabolic specialization of astrocytes that depends on local neuron-astrocyte circuits. In the last version we considered the mitochondrial DNA results adds a significant heuristic value to the manuscript and set up the basis for future research on the mechanisms of activity-dependent astrocyte heterogeneity.